# Towards Understanding the Transferability of Deep Representations

## Abstract

Deep neural networks trained on a wide range of datasets demonstrate impressive transferability. Deep features appear general in that they are applicable to many datasets and tasks. Such property is in prevalent use in real-world applications. A neural network pretrained on large datasets, such as ImageNet, can significantly boost generalization and accelerate training if fine-tuned to a smaller target dataset. Despite its pervasiveness, few effort has been devoted to uncovering the reason of transferability in deep feature representations. This paper tries to understand transferability from the perspectives of improved generalization, optimization and the feasibility of transferability. We demonstrate that 1) Transferred models tend to find flatter minima, since their weight matrices stay close to the original flat region of pretrained parameters when transferred to a similar target dataset; 2) Transferred representations make the loss landscape more favorable with improved Lipschitzness, which accelerates and stabilizes training substantially. The improvement largely attributes to the fact that the principal component of gradient is suppressed in the pretrained parameters, thus stabilizing the magnitude of gradient in backpropagation. 3) The feasibility of transferability is related to the similarity of both input and label. And a surprising discovery is that the feasibility is also impacted by the training stages in that the transferability first increases during training, and then declines. We further provide a theoretical analysis to verify our observations.

## 1 Introduction

The last decade has witnessed the enormous success of deep neural networks in a wide range of applications. Deep learning has made unprecedented advances in many research fields, including computer vision, natural language processing, and robotics. Such great achievement largely attributes to several desirable properties of deep neural networks. One of the most prominent properties is the transferability of deep feature representations.

Transferability is basically the desirable phenomenon that deep feature representations learned from one dataset can benefit optimization and generalization on different datasets or even different tasks, e.g. from real images to synthesized images, and from image recognition to object detection (Yosinski et al., 2014). This is essentially different from traditional learning techniques and is often regarded as one of the parallels between deep neural networks and human learning mechanisms.

In real-world applications, practitioners harness transferability to overcome various difficulties. Deep networks pretrained on large datasets are in prevalent use as general-purpose feature extractors for downstream tasks (Donahue et al., 2014). For small datasets, a standard practice is to fine-tune a model transferred from large-scale dataset such as ImageNet (Russakovsky et al., 2015) to avoid over-fitting. For complicated tasks such as object detection, semantic segmentation and landmark localization, ImageNet pretrained networks accelerate training process substantially (Oquab et al., 2014; He et al., 2018). In the NLP field, advances in unsupervised pretrained representations have enabled remarkable improvement in downstream tasks (Vaswani et al., 2017; Devlin et al., 2019).

Despite its practical success, few efforts have been devoted to uncovering the underlying mechanism of transferability. Intuitively, deep neural networks are capable of preserving the knowledge learned on one dataset after training on another similar dataset (Yosinski et al., 2014; Li et al., 2018b; 2019). This is even true for notably different datasets or apparently different tasks. Another line of works have observed several detailed phenomena in the transfer learning of deep networks (Kirkpatrick

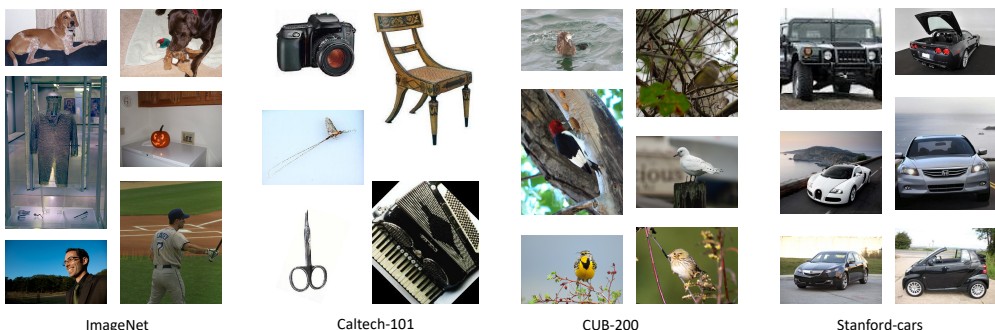

Figure 1: Examples of object recognition datasets. Caltech-101 (Fei-Fei et al., 2004) is more similar to ImageNet with 101 classes of common objects. CUB-200 (Welinder et al., 2010) and Stanford Cars (Krause et al., 2013) are essentially different with various kinds of birds and vehicles, respectively.

et al., 2016; Kornblith et al., 2019), yet it remains unclear why and how the transferred representations are beneficial to the generalization and optimization perspectives of deep networks.

The present study addresses this important problem from several new perspectives. We first probe into how pretrained knowledge benefits *generalization*. Results indicate that models fine-tuned on target datasets similar to the pretrained dataset tend to stay close to the transferred parameters. In this sense, transferring from a similar dataset makes fine-tuned parameters stay in the flat region around the pretrained parameters, leading to flatter minima than training from scratch.

Another key to transferability is that transferred features make the optimization landscape significantly improved with better Lipschitzness, which eases *optimization*. Results show that the landscapes with transferred features are smoother and more predictable, fundamentally stabilizing and accelerating training especially at the early stages of training. This is further enhanced by the proper scaling of gradient in back-propagation. The principal component of gradient is suppressed in the transferred weight matrices, controlling the magnitude of gradient and smoothing the loss landscapes.

We also investigate a common concern raised by practitioners: *when* is transfer learning helpful to target tasks? We test the transferability of pretrained networks with varying inputs and labels. Instead of the similarity between pretrained and target inputs, what really matters is the similarity between the pretrained and target tasks, i.e. both inputs and labels are required to be sufficiently similar. We also investigate the relationship between pretraining epoch and transferability. Surprisingly, although accuracy on the pretrained dataset increases throughout training, transferability first increases at the beginning and then decreases significantly as pretraining proceeds.

Finally, this paper gives a theoretical analysis based on two-layer fully connected networks. Theoretical results consistently justify our empirical discoveries. The analysis here also casts light on deeper networks. We believe the mechanism of transferability is the fundamental property of deep neural networks and the in-depth understanding presented here may stimulate further algorithmic advances.

## 2 RELATED WORK

There exists extensive literature on transferring pretrained representations to learn an accurate model on a target dataset. Donahue et al. (2014) employed a brand-new label predictor to classify features extracted by the pre-trained feature extractor at different layers of AlexNet (Krizhevsky et al., 2012). Oquab et al. (2014) showed deep features can benefit object detection tasks despite the fact that they are trained for image classification. Ge & Yu (2017) introduced a selective joint fine-tuning scheme for improving the performance of deep learning tasks under the scenario of insufficient training data.

The enormous success of the transferability of deep networks in applications stimulates empirical studies on fine-tuning and transferability. Yosinski et al. (2014) observed the transferability of deep feature representations decreases as the discrepancy between pretrained task and target task increases and gets worse in higher layers. Another phenomenon of catastrophic forgetting as discovered by Kirkpatrick et al. (2016) describes the loss of pretrained knowledge when fitting to distant tasks. Huh et al. (2016) delved into the influence of ImageNet pretrained features by pretraining on various

subsets of the ImageNet dataset. Kornblith et al. (2019) further demonstrated that deep models with better ImageNet pretraining performance can transfer better to target tasks.

As for the techniques used in our analysis, Li et al. (2018a) proposed the impact of the scaling of weight matrices on the visualization of loss landscapes. Santurkar et al. (2018) proposed to measure the variation of loss to demonstrate the stability of loss function. Du et al. (2019) provided a powerful framework of analyzing two-layer over-parametrized neural networks, with elegant results and no strong assumptions on input distributions, which is flexible for our extensions to transfer learning.

## 3 Transferred Knowledge Induces Better Generalization

A basic observation of transferability is that tasks on target datasets more similar to the pretrained dataset have better performance. We delve deeper into this phenomenon by experimenting on a variety of target datasets (Figure 1), carried out with two common settings: 1) train only the last layer by *fixing* the pretrained network as the feature extractor and 2) train the whole network by *fine-tuning* from the pretrained representations. Results in Table 1 clearly demonstrate that, for both settings and for all target datasets, the training error converges to nearly zero while the generalization error varies significantly. In particular, a network pretrained on more similar dataset tends to generalize better and converge faster on the target dataset. A natural implication is that the knowledge learned from the pretrained networks can only be preserved to different extents for different target datasets.

Table 1: Transferring to different datasets with ImageNet pretrained networks fixed or fine-tuned.

| Dataset | training error (fixed) | test error (fixed) | $\frac{1}{\sqrt{n}}\|\mathbf{W} - \mathbf{W}_0\|_F$ | training error (fine-tuned) | test error (fine-tuned) | $\frac{1}{\sqrt{n}}\sum_l \|\mathbf{W}_{(l)} - \mathbf{W}_{0(l)}\|_F$ |
|---|---|---|---|---|---|---|
| Webcam | 0.00±0 | 0.45±0.05 | 0.096±0.007 | 0.00±0 | 0.45±0.09 | 0.94±0.10 |
| Stanford Cars | 0.00±0 | 24.2±0.73 | 0.165±0.003 | 0.00±0 | 13.95±0.44 | 3.70±0.26 |
| Caltech-101 | 0.00±0 | 6.24±0.37 | 0.059±0.005 | 0.00±0 | 4.57±0.40 | 1.22±0.21 |
| Synthetic | 0.03±0.01 | 0.81±0.19 | 0.015±0.003 | 0.00±0 | 0.75±0.11 | 0.65±0.12 |
| CUB-200 | 0.15±0.03 | 35.10±0.50 | 0.262±0.006 | 0.04±0.01 | 21.04±0.36 | 2.38±0.33 |

We substantiate this implication with the following experiments. To analyze to what extent the knowledge learned from pretrained dataset is preserved, for the *fixing* setting, we compute the Frobenius norm of the deviation between fine-tuned weight $\mathbf{W}$ and pretrained weight $\mathbf{W}_0$ as $\frac{1}{\sqrt{n}}\|\mathbf{W} - \mathbf{W}_0\|_F$, where $n$ denotes the number of target examples (for the *fine-tuning* setting, we compute the sum of deviations in all layers $\frac{1}{\sqrt{n}}\sum_l \|\mathbf{W}_{(l)} - \mathbf{W}_{0(l)}\|_F$). Results are shown in Figure 2. It is surprising that although accuracy may oscillate, $\frac{1}{\sqrt{n}}\|\mathbf{W} - \mathbf{W}_0\|_F$ increases monotonously throughout the training process and eventually converges. Datasets with larger visual similarity to ImageNet have smaller deviations from pretrained weight, and $\frac{1}{\sqrt{n}}\|\mathbf{W} - \mathbf{W}_0\|_F$ also converges faster. Another observation is that $\frac{1}{\sqrt{n}}\|\mathbf{W} - \mathbf{W}_0\|_F$ is approximately proportional to the generalization error shown in Table 1.

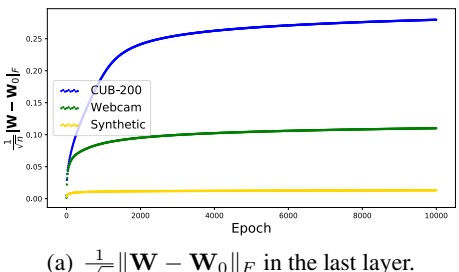
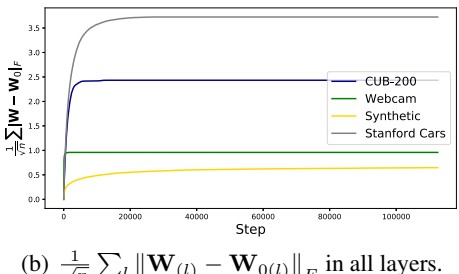

(a) $\frac{1}{\sqrt{n}}\|\mathbf{W} - \mathbf{W}_0\|_F$ in the last layer.  (b) $\frac{1}{\sqrt{n}}\sum_l \|\mathbf{W}_{(l)} - \mathbf{W}_{0(l)}\|_F$ in all layers.

Figure 2: The deviation of the weight parameters from the pretrained ones in the transfer process to different target datasets. For all datasets, $\frac{1}{\sqrt{n}}\|\mathbf{W} - \mathbf{W}_0\|_F$ increases monotonously. More knowledge can be preserved on target datasets more similar to ImageNet, yielding smaller $\frac{1}{\sqrt{n}}\|\mathbf{W} - \mathbf{W}_0\|_F$.

Why is preserving pretrained knowledge related to better generalization? From the experiments above, we can observe that models preserving more transferred knowledge (i.e. yielding smaller $\frac{1}{\sqrt{n}}\|\mathbf{W}-\mathbf{W}_0\|_F$) generalize better. It is reasonable to hypothesize that $\frac{1}{\sqrt{n}}\|\mathbf{W}-\mathbf{W}_0\|_F$ is implicitly bounded in the transfer process, and that the bound is related to the similarity between pretrained and target datasets (We will formally study this conjecture in the theoretical analysis). Intuitively, a neural network attempts to fit the training data by twisting itself from the initialization point. For similar datasets the twist will be mild, with the weight parameters staying closer to the pretrained parameters.

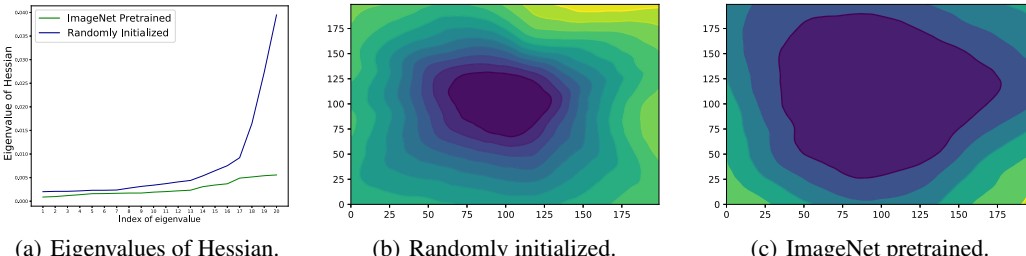

(a) Eigenvalues of Hessian.    (b) Randomly initialized.    (c) ImageNet pretrained.

Figure 3: Generalization w.r.t. landscape at *convergence* on Stanford Cars dataset with ResNet-50. (a) Comparison of the top 20 eigenvalues of the Hessian matrix. ImageNet pretrained networks have smaller eigenvalues, indicating flatter minima. (b) (c) Comparison of landscapes centered at the minima. We use the filter normalization (Li et al., 2018a) to avoid the impact of weight scale. Randomly initialized networks end up with sharper minima. Pretrained networks stay in flat regions.

Such property of staying near the pretrained weight is crucial for understanding the improvement of generalization. Since optimizing deep networks inevitably runs into local minima, a common belief of deep networks is that the optimization trajectories of weight parameters on different datasets will be essentially different, leading to distant local minima. To justify whether this is true, we compare the weight matrices of training from scratch and using ImageNet pretrained representations in Figure 4. Results are quite counterintuitive. The local minima of different datasets using ImageNet pretraining are closed to each other, all concentrating around ImageNet pretrained weight. However, the local minima of training from scratch and ImageNet pretraining are way distant, even on the same dataset.

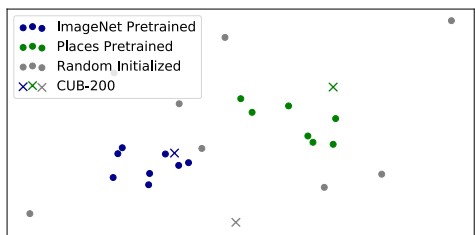

Figure 4: The t-SNE (van der Maaten & Hinton, 2008) visualization of the weight matrices of the pretrained and randomly initialized networks on different datasets. Surprisingly, weight matrices on the same dataset may be distant at *convergence* when using different initializations. On the contrary, even for discrepant datasets, the weight matrices stay close to the initialization when using the same pretrained parameters.

This provides us with a clear picture of how transferred representations improve generalization on target datasets. Rich studies have indicated that the properties of local minima are directly related to generalization (Keskar et al., 2017; Izmailov et al., 2018). Using pretrained representations restricts weight matrices to stay near the pretrained weight. Since the pretrained dataset is usually sufficiently large and of high-quality, transferring their representations will lead to *flatter* minima located in large flat basins. On the contrary, training from scratch may find sharper minima. To observe this, we adopt filter normalization (Li et al., 2018a) as the visualization tool, and illustrate the loss landscapes around the minima in Figure 3. This observation concurs well with the experiments above. The weight matrices for datasets similar to pretrained ones deviate less from pretrained weights and stay in the flat region. On more different datasets, the weight matrices have to go further from pretrained weights to fit the data and may run out of the flat region.

# 4 PROPERLY PRETRAINED NETWORKS ENABLE BETTER LOSS LANDSCAPES

A common belief of modern deep networks is the improvement of loss landscapes with techniques such as BatchNorm (Ioffe & Szegedy, 2015) and residual structures (He et al., 2016). Li et al. (2018a); Santurkar et al. (2018) validated this improvement when the model is close to convergence. However, it is often overlooked that loss landscapes can still be messy at the *initialization* point. To verify this conjecture, we visualize the loss landscapes centered at the initialization point of the 25th layer of ResNet-50 in Figure 5. (Visualizations of the other layers can be found in Appendix B.4.) ImageNet pretrained networks have much smoother landscape than networks trained with random initialization. The improvement of loss landscapes at the initialization point directly gives rise to the acceleration of training. Concretely, transferred features help ameliorate the chaos of loss landscape with improved Lipschitzness in the early stages of training. Thus, gradient-based optimization method can easily escape from the initial region where the loss is very large.

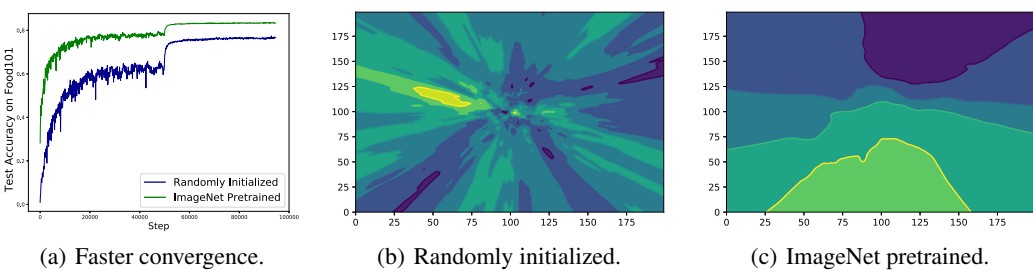

| (a) Faster convergence. | (b) Randomly initialized. | (c) ImageNet pretrained. |

Figure 5: Optimization landscape at *initialization*. (a) A figure of convergence on fine-grained classification task Food-101. The convergence of fine-tuning from pretrained networks is significantly faster than training from scratch. (b) (c) Visualizations of loss landscapes of the 25th layer in ResNet-50 centered at initialization on Stanford Cars dataset. ImageNet pretrained landscape is much smoother, indicating better Lipschitzness and predictability of the loss function. Randomly initialized landscape is more bumpy, making optimization unstable and inefficient.

The properties of loss landscapes influence the optimization fundamentally. In randomly initialized networks, going in the direction of gradient may lead to large variation in the loss function. On the contrary, ImageNet pretrained features make the geometry of loss landscape much more predictable, and a step in gradient direction will lead to mild decrease of loss function. To demonstrate the impact of transferred features on the stability of loss function, we further analyze the variation of loss in the direction of gradient in Figure 6. For each step in the training process, we compute the gradient of the loss and measure how the loss changes as we move the weight matrix in that direction. We can clearly observe that in contrast to networks with transferred features, randomly initialized networks have larger variation along the gradient, where a step along the gradient leads to drastic change in the loss. Why can transferred features control the magnitude of gradient and smooth the loss landscape?

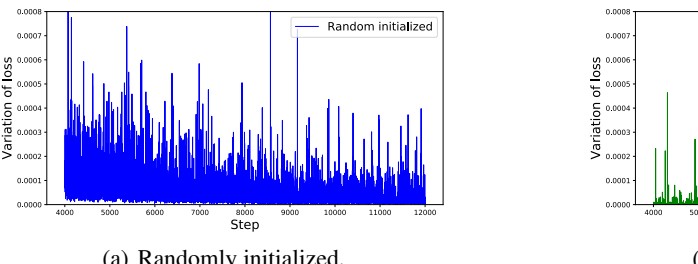

| (a) Randomly initialized. | (b) ImageNet pretrained. |

Figure 6: Variation of the loss in ResNet-50 with ImageNet pretrained weight and random initialization. We compare the variation of loss function in the direction of gradient during the training process on CUB-200 dataset. The variation of pretrained networks is substantially smaller than the randomly initialized one, implying a more desirable loss landscape and more stable optimization.

A natural explanation is that transferred weight matrices provide appropriate transform of gradient

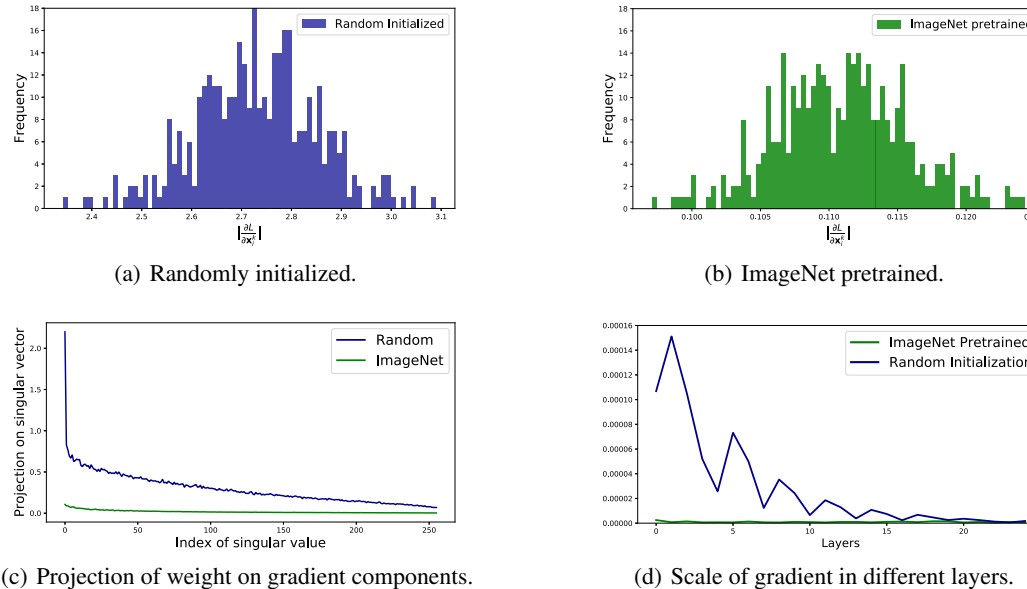

(a) Randomly initialized.

(b) ImageNet pretrained.

(c) Projection of weight on gradient components.

(d) Scale of gradient in different layers.

Figure 7: The stabilization of gradient by pretrained weights on CUB-200. (a) (b) Distribution of the magnitude of gradient in the 25th layer of ResNet-50. (c) Magnitude of the projection of weight on the singular vectors of gradient. (d) Scaling of the gradient in different layers in back-propagation.

in each layer and help stabilize its magnitude. Note that in deep neural networks, the gradient w.r.t. each layer is computed through back-propagation by $\frac{\partial L}{\partial \mathbf{x}_i^{k-1}} = \mathbf{W}_k \mathbb{I}_i^k \left( \frac{\partial L}{\partial \mathbf{x}_i^k} \right)$, where $\mathbb{I}_i^k$ denotes the activation of $\mathbf{x}_i$ at layer $k$. The weight matrices $\mathbf{W}_k$ function as the *scaling* factor of gradient in back-propagation. Basically, a randomly initialized weight matrix will multiply the magnitude of gradient by its norm. In pretrained weight matrices, situation is completely different. To delve into this, we decompose the gradient into singular vectors and measure the projections of weight matrices in these principal directions. Results are shown in Figure 7(c). During pretraining, the singular vectors of the gradient with large singular values are shrunk in the weight matrices. Thus, the magnitude of gradient back-propagated through a pretrained layer is controlled. In this sense, pretrained weight matrices stabilize the magnitude of gradient especially in lower layers. We visualize the magnitude and scaling of gradient of different layers in ResNet-50 in Figure 7. The gradient of randomly initialized networks grows fast with layer numbers during back-propagation while the gradient of ImageNet pretrained networks remains stable. Note that ResNet-50 already incorporates BatchNorm and skip-connections to improve the gradient flow, and pretrained representations can stabilize the magnitude of gradient substantially even in these modern networks. We complete this analysis by visualizing the change of landscapes during back-propagation in Section B.4.

## 5 WHEN IS TRANSFER LEARNING FEASIBLE IN DEEP NETWORKS?

Transferring from pretrained representations boosts performance in a wide range of applications. However, as discovered by He et al. (2018); Kornblith et al. (2019), there still exist cases when pretrained representations provide no help for target tasks or even downgrade test accuracy. Hence, the conditions on which transfer learning is feasible is an important open problem to be explored. In this section, we delve into the feasibility of transfer learning with extensive experiments, while the theoretical perspectives are presented in the next section. We hope our analysis will provide insights into how to adopt transfer learning by practitioners.

### 5.1 CHOICES OF A PRETRAINED DATASET

As a common practice, people choose datasets similar to the target dataset for pretraining. However, how can we determine whether a dataset is sufficiently similar to a target dataset? We verify with experiments that the similarity depends on the nature of tasks, i.e. both inputs and labels matter.

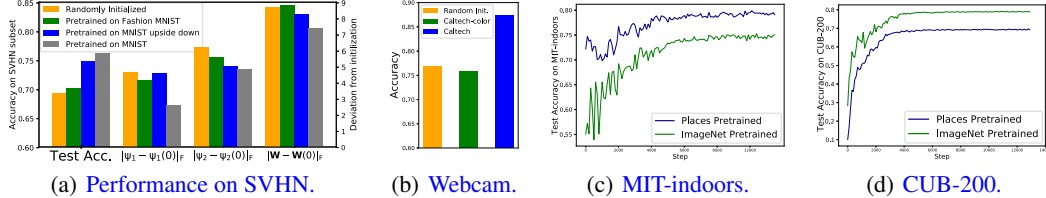

(a) Performance on SVHN.  (b) Webcam.  (c) MIT-indoors.  (d) CUB-200.

Figure 8: Performance by varying either the inputs or labels. (a) Accuracy and deviation of transferring from different datasets to MNIST, where $\psi_1$ and $\psi_2$ denote the convolutional kernels in the first and the second layers, and $\mathbf{W}$ denotes the weight of the fully-connected layer. (b) Accuracy on Webcam with Caltech and Caltech-color pretraining. (c) (d) Test accuracy and convergence of transferring to MIT-indoors and CUB-200 from models pretrained on large-scale ImageNet and Places, respectively.

**Varying input with fixed labels.** We randomly sample 600 images from the original SVHN dataset, and fine-tune the MNIST pretrained LeNet (LeCun et al., 1998) to this SVHN subset. For comparison, we pretrain other two models on MNIST with images upside down and Fashion-MNIST (Xiao et al., 2017), respectively. Note that for all three pretrained models, the dataset sizes, labels, and the number of images per class are kept exactly the same, and thus the only difference lies in the image pixels themselves. Results are shown in Figure 8(a). Compared to training from scratch, MNIST pretrained features improve generalization significantly. Upside-down MNIST shows slightly worse generalization performance than the original one. In contrast, fine-tuning from Fashion-MNIST barely improves generalization. We also compute the deviation from pretrained weight of each layer. The weight matrices and convolutional kernel deviation of Fashion-MNIST pretraining show no improvement over training from scratch. A reasonable implication here is that choosing a model pretrained on a more similar dataset in the inputs yields a larger performance gain.

**Varying labels with fixed input.** We train a ResNet-50 model on Caltech-101 and then fine-tune it to Webcam (Saenko et al., 2010). We train another ResNet-50 to recognize the *color* of the upper part of Caltech-101 images and fine-tune it to Webcam. Results in Figure 8(b) indicate that the latter one provides no improvement over training on Webcam from scratch, while pretraining on standard Caltech-101 significantly boosts performance. Models generalizing very well on similar images are not transferable to the target dataset with totally different labels. These experiments challenge the common perspective of *similarity* between datasets. The description of similarity using the input (images) themselves is just one point. Another key factor of similarity is the relationship between the nature of tasks (labels). This observation is further in line with our theoretical analysis in Section 6.

Similar results are observed on datasets of larger scale for the above two cases (See Figure 8(c)8(d)). We fine-tune deep representations pretrained on ImageNet and Places (Zhou et al., 2018) (a large-scale dataset of scene understanding) to CUB-200 and MIT-indoors datasets. As a fine-grained object recognition task, CUB-200 benefits more from ImageNet pretrained features. In contrast, MIT-indoors is a scene dataset more similar to Places, thus it benefits more from Places pretraining.

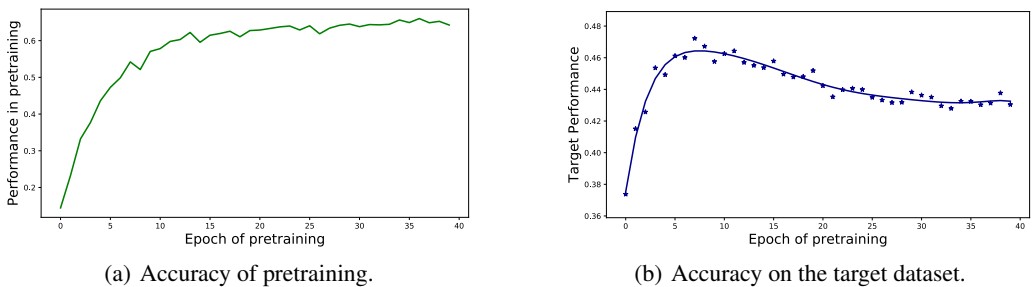

(a) Accuracy of pretraining.  (b) Accuracy on the target dataset.

Figure 9: Transfer performance w.r.t. the pretraining epochs from Food-101 to CUB-200. (a) Accuracy of pretraining task and (b) Accuracy on target dataset by fine-tuning from models pretrained for different numbers of epochs. Although accuracy of pretraining increases stably, transferability increases only during the early epochs and decreases afterwards.

## 5.2 Choices of Pretraining Epochs

Currently, people usually train a model on ImageNet until it converges and use it as the pretrained parameters. However, the final model do not necessarily have the highest transferability. To see this, we pretrain a ResNet-50 model on Food-101 (Bossard et al., 2014) and transfer it to CUB-200, with results shown in Figure 9. During the early epochs, the transferability increases sharply. As we continue pretraining, although the test accuracy on the pretraining dataset continues increasing, the test accuracy on the target dataset starts to decline, indicating downgraded transferability. Intuitively, during the early epochs, the model learns general knowledge that is informative to many datasets. As training goes on, however, the model starts to fit the specific knowledge of the pretrained dataset and even fit noise. Such dataset-specific knowledge is usually detrimental to the transfer performance. This interesting finding implies a promising direction for improving the *de facto* pretraining method: Instead of seeking for a model with higher accuracy only on the pretraining dataset, a more transferable model can be pretrained with appropriate epochs such that the fine-tuning accuracies on a diverse set of target tasks are advantageous. Algorithms for pretraining should take this point into consideration.

## 6 Theoretical Analysis

We have shown through extensive empirical analysis that transferred features exert a fundamental impact on generalization and optimization performance, and provided some insights for the feasibility of transfer learning. In this section, we analyze some of our empirical observations from a theoretical perspective. We base our analysis on two-layer fully connected networks with ReLU activation and sufficiently many hidden units. Our theoretical results are in line with the experimental findings.

### 6.1 Setup

Denote by $\sigma(\cdot)$ the ReLU activation function, $\sigma(z) = \max\{z, 0\}$. $\mathbb{I}\{A\}$ is the indicator function, i.e. $\mathbb{I}\{A\} = 1$ if $A$ is true and $0$ otherwise. $[m]$ is the set of integers ranging from 1 to $m$. Consider a two-layer ReLU network of $m$ hidden units $f_{\mathbf{W},\mathbf{a}}(\mathbf{x}) = \frac{1}{\sqrt{m}}\mathbf{a}^\top \sigma(\mathbf{W}^\top \mathbf{x})$, with $\mathbf{x} \in \mathbb{R}^d$ as input and $\mathbf{W} = (\mathbf{w}_1, \cdots, \mathbf{w}_m) \in \mathbb{R}^{d\times m}$ as the weight matrix. We are provided with $n_Q$ samples $\{\mathbf{x}_{Q,i}, y_{Q,i}\}_{i=1}^{n_Q}$ drawn i.i.d. from the target distribution $Q$ as the target dataset and a weight matrix $\mathbf{W}(P)$ pretrained on $n_P$ samples $\{\mathbf{x}_{P,i}, y_{P,i}\}_{i=1}^{n_P}$ drawn i.i.d. from pretrained distribution $P$. Suppose $\|\mathbf{x}\|_2 = 1$ and $|y| \leq 1$. Our goal is transferring the pretrained $\mathbf{W}(P)$ to learn an accurate model $\mathbf{W}(Q)$ for the target distribution $Q$. When training the model on the pretraining dataset, we initialize the weight as: $\mathbf{w}_r(0) \sim \mathcal{N}(\mathbf{0}, \kappa^2\mathbf{I})$, $a_r \sim$ unif $(\{-1, 1\})$, where $\forall r \in [m]$ and $\kappa$ is a constant.

For both pretraining and fine-tuning, the objective function of the model is the squared loss $L(\mathbf{W}) = \frac{1}{2}(\mathbf{y} - f_{\mathbf{W},\mathbf{a}}(\mathbf{X}))^\top(\mathbf{y} - f_{\mathbf{W},\mathbf{a}}(\mathbf{X}))$. Note that $\mathbf{a}$ is fixed throughout training and $\mathbf{W}$ is updated with gradient descent. The learning rate is set to $\eta$.

We base our analysis on the theoretical framework of Du et al. (2019), since it provides elegant results on convergence of two-layer ReLU networks without strong assumptions on the input distributions, facilitating our extension to the transfer learning scenarios. In our analysis, we use the Gram matrices $\mathbf{H}_P^\infty \in \mathbb{R}^{n_P \times n_P}$ and $\mathbf{H}_Q^\infty \in \mathbb{R}^{n_Q \times n_Q}$ to measure the quality of pretrained input and target input as

$$\mathbf{H}_{P,ij}^\infty = \mathbb{E}_{\mathbf{w}\sim\mathcal{N}(\mathbf{0},\mathbf{I})}[\mathbf{x}_{P,i}^\top\mathbf{x}_{P,j}\mathbb{I}\{\mathbf{w}^\top\mathbf{x}_{P,i} \geq 0, \mathbf{w}^\top\mathbf{x}_{P,j} \geq 0\}] = \frac{\mathbf{x}_{P,i}^\top\mathbf{x}_{P,j}(\pi - \arccos(\mathbf{x}_{P,i}^\top\mathbf{x}_{P,j}))}{2\pi},$$

$$\mathbf{H}_{Q,ij}^\infty = \mathbb{E}_{\mathbf{w}\sim\mathcal{N}(\mathbf{0},\mathbf{I})}[\mathbf{x}_{Q,i}^\top\mathbf{x}_{Q,j}\mathbb{I}\{\mathbf{w}^\top\mathbf{x}_{Q,i} \geq 0, \mathbf{w}^\top\mathbf{x}_{Q,j} \geq 0\}] = \frac{\mathbf{x}_{Q,i}^\top\mathbf{x}_{Q,j}(\pi - \arccos(\mathbf{x}_{Q,i}^\top\mathbf{x}_{Q,j}))}{2\pi}.$$

To quantify the relationship between pretrained input and target input, we define the following Gram matrix $\mathbf{H}_{PQ}^\infty \in \mathbb{R}^{n_P \times n_Q}$ across samples drawn from $P$ and $Q$:

$$\mathbf{H}_{PQ,ij}^\infty = \mathbb{E}_{\mathbf{w}\sim\mathcal{N}(\mathbf{0},\mathbf{I})}[\mathbf{x}_{P,i}^\top\mathbf{x}_{Q,j}\mathbb{I}\{\mathbf{w}^\top\mathbf{x}_{P,i} \geq 0, \mathbf{w}^\top\mathbf{x}_{Q,j} \geq 0\}] = \frac{\mathbf{x}_{P,i}^\top\mathbf{x}_{Q,j}(\pi - \arccos(\mathbf{x}_{P,i}^\top\mathbf{x}_{Q,j}))}{2\pi}.$$

Assume Gram matrices $\mathbf{H}_P^\infty$ and $\mathbf{H}_Q^\infty$ are invertible with smallest eigenvalue $\lambda_P$ and $\lambda_Q$ greater than zero. $\mathbf{H}_P^{\infty-1}\mathbf{y}_P$ characterizes the labeling function of pretrained tasks. $\mathbf{y}_{P\to Q} \triangleq \mathbf{H}_{PQ}^{\infty\top}\mathbf{H}_P^{\infty-1}\mathbf{y}_P$ further transforms the pretrained labeling function to the target labels. A critical point in our analysis is $\mathbf{y}_Q - \mathbf{y}_{P\to Q}$, which measures the *task similarity* between target label and transformed label.

## 6.2 Improved Lipschitzness of Loss Function

To analyze the Lipschitzness of loss function, a reasonable objective is the magnitude of gradient, which is a direct manifestation of the Lipschitz constant. We analyze the gradient w.r.t. the activations. For the magnitude of gradient w.r.t. the activations, we show that the Lipschitz constant is significantly reduced when the pretrained and target datasets are similar in both inputs and labels.

**Theorem 1** (**The effect of transferred features on the Lipschitzness of the loss**). *Denote by $\mathbf{X}^1$ the activations in the target dataset. For a two-layer networks with sufficiently large number of hidden unit $m$ defined in Section 6.1, if $m \geq \text{poly}(n_P, n_Q, \delta^{-1}, \lambda_P^{-1}, \lambda_Q^{-1}, \kappa^{-1})$, $\kappa = O\left(\frac{\lambda_P^2 \delta}{n_P^2 n_Q^{\frac{1}{2}}}\right)$, with probability no less than $1 - \delta$ over the random initialization,*

$$\|\frac{\partial L(\mathbf{W}(P))}{\partial \mathbf{X}^1}\|^2 = \|\frac{\partial L(\mathbf{W}(0))}{\partial \mathbf{X}^1}\|^2 - \mathbf{y}_Q^\top \mathbf{y}_Q + (\mathbf{y}_Q - \mathbf{y}_{P \to Q})^\top (\mathbf{y}_Q - \mathbf{y}_{P \to Q})$$
$$+ \frac{\text{poly}(n_P, n_Q, \delta^{-1}, \lambda_P^{-1}, \kappa^{-1})}{m^{\frac{1}{4}}} + O\left(\frac{n_P^2 n_Q^{\frac{1}{2}} \kappa}{\lambda_P^2 \delta}\right). \tag{1}$$

This provides us with theoretical explanation of experimental results in Section 4. The control of Lipschitz constant relies on the similarity between tasks in both input and labels. If the original target label is similar to the label transformed from the pretrained label, i.e. $\|\mathbf{y}_Q - \mathbf{y}_{P \to Q}\|_2^2$ is small, the Lipschitzness of loss function will be significantly improved. On the contrary, if the pretrained and target tasks are completely different, the transformed label will be discrepant with target label, resulting in larger Lipschitz constant of the loss function and worse landscape in the fine-tuned model.

## 6.3 Improved Generalization

Recall in Section 3 that we have investigated the weight change $\|\mathbf{W}(Q) - \mathbf{W}(P)\|_F$ during training and point out the role it plays in understanding the generalization. In this section, we show that $\|\mathbf{W}(Q) - \mathbf{W}(P)\|_F$ can be bounded with terms depicting the similarity between pretrained and target tasks. Note that the Rademacher complexity of the function class is bounded with $\|\mathbf{W}(Q) - \mathbf{W}(P)\|_F$ as shown in the seminal work (Arora et al., 2019), thus the generalization error is directly related to $\|\mathbf{W}(Q) - \mathbf{W}(P)\|_F$. We still use the Gram matrices defined in Section 6.1.

**Theorem 2** (**The effect of transferred features on the generalization error**). *For a two-layer networks with $m \geq \text{poly}(n_P, n_Q, \delta^{-1}, \lambda_P^{-1}, \lambda_Q^{-1}, \kappa^{-1})$, $\kappa = O\left(\frac{\lambda_P^2 \lambda_Q^2 \delta}{n_P^2 n_Q^{\frac{1}{2}}}\right)$, with probability no less than $1 - \delta$ over the random initialization,*

$$\|\mathbf{W}(Q) - \mathbf{W}(P)\|_F \leq \sqrt{(\mathbf{y}_Q - \mathbf{y}_{P \to Q})^\top \mathbf{H}_Q^{\infty -1} (\mathbf{y}_Q - \mathbf{y}_{P \to Q})}$$
$$+ O\left(\frac{n_P n_Q^{\frac{1}{2}} \kappa^{\frac{1}{2}}}{\lambda_P \lambda_Q \delta^{\frac{1}{2}}}\right) + \frac{\text{poly}(n_P, n_Q, \delta^{-1}, \lambda_P^{-1}, \lambda_Q^{-1}, \kappa^{-1})}{m^{\frac{1}{4}}}. \tag{2}$$

This result is directly related to the generalization error and casts light on our experiments in Section 5.1. Note that when training on the target dataset from scratch, the upper bound of $\|\mathbf{W}(Q) - \mathbf{W}(0)\|_F$ is $\mathbf{y}_Q^\top \mathbf{H}_Q^{\infty -1} \mathbf{y}_Q$. By fine-tuning from a similar pretrained dataset where the transformed label is close to target label, the generalization error of the function class is hopefully reduced. On the contrary, features pretrained on discrepant tasks do not transfer to classification task in spite of similar images since they have disparate labeling functions. Another example is fine-tuning to Food-101 as in the experiment of Kornblith et al. (2019). Since it is a fine-grained dataset with many similar images, $\mathbf{H}_Q^\infty$ will be more singular than common tasks, resulting in a larger deviation from the pretrained weight. Hence even transferring from ImageNet, the performance on Food-101 is still far from satisfactory.

## 7 Conclusion: Behind Transferability

Why are deep representations pretrained from modern neural networks generally transferable to novel tasks? When is transfer learning feasible enough to consistently improve the target task performance?

These are the key questions in the way of understanding modern neural networks and applying them to a variety of real tasks. This paper performs the first in-depth analysis of the transferability of deep representations from both empirical and theoretical perspectives. The results reveal that pretrained representations will improve both generalization and optimization performance of a target network provided that the pretrained and target datasets are sufficiently similar in both input and labels. With this paper, we show that transfer learning, as an initialization technique of neural networks, exerts implicit regularization to restrict the networks from escaping the flat region of pretrained landscape.

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

# A IMPLEMENTATION DETAILS

In this section, we provide details of the architectures, setup, methods of visualizations in our analysis. The codes and visualizations are attached with the submission and will be made available online.

**Models.** We implement all models on PyTorch with 2080Ti GPUs. For object recognition and scene recognition tasks, we use standard ResNet-50 from torchvision. ImageNet pretrained models can be found in torchvision, and Places pretrained models are provided by Zhou et al. (2018). During fine-tuning we use a batch size of 32 and set the initial learning rate to 0.01 with 0.9 momentum following the protocol of (Li et al., 2018b). For fine-tuning, we train the model for 200 epochs. We decay the learning rate by 0.1 with the time of decay set by cross validation. In Figure 2(a) where the pretrained ResNet-50 functions as feature extractor, the downstream classifier is a two-layer ReLU network with Batch-Norm and Leaky-ReLU non-linearity. The number of hidden unit is 512. For this task, the backbone ResNet-50 is fixed, with the downstream two-layer classifier trained with momentum SGD. The learning rate is set to 0.01 with 0.9 momentum, and remains constant throughout training.

For digit recognition tasks, we use LeNet (LeCun et al., 1998). The learning rate is also set to 0.01, with $5 \times 10^{-4}$ weight decay. The batch-size is set to 64. We train the model for 100 epochs.

**Fine-tuning.** We follow the protocol of fine-tuning as in the previous paragraphs. In Tables 1 and 2, we run all the experiments for 3 times and report their mean and variance. For Table 2, the improvement of fine-tuing is calculated with the generalization error of fine-tuning divided by the generalization error of training from scratch.

**Visualization of loss landscapes.** We use techniques similar to filter normalization to provide an accurate analysis of loss landscapes (Li et al., 2018a). Note that ReLU networks are invariant to the scaling of weight parameters. To remove this scaling effect, the direction used in visualization should be normalized in a filter-wise way. Concretely, the axes of each landscape figure are two random Gaussian orthogonal vectors normalized by the scale of each filter in the convolutional layers. Concretely, suppose the parameter of the center point is $\theta$. $\theta_{i,j}$ denotes the $j$-th filter of the $i$-th layer. Suppose the two unit orthogonal vectors are $a$ and $b$. Then with filter normalization, $a_{i,j} \leftarrow \frac{a_{i,j}}{\|a_{i,j}\|}\|\theta_{i,j}\|$ and $b_{i,j} \leftarrow \frac{b_{i,j}}{\|b_{i,j}\|}\|\theta_{i,j}\|$. For each point (i.e. pixel) $(p,q)$ in the plot, the value is evaluated with $g(p,q) = L(f(\theta + \eta(pa + qb)))$, where $L$ denotes the loss function, $f$ denotes the neural networks. $\eta$ is a parameter to control the scale of the plot. In all visualization images of ResNet-50, the resolution is $200 \times 200$, i.e. $p = -100, -99, \cdots, 98, 99$ and $q = -100, -99, \cdots, 98, 99$. For additional details of filter normalization, please refer to (Li et al., 2018a). $\eta$ is set to 0.001, which is of the same order as 10 times the step size in training. This is a reasonable scale if we want to study the local loss landscape of model using SGD. For fair comparison between the pretrained landscapes and randomly initialized landscapes, the scale of loss variation in each plot is exactly the same. The difference of loss value between each contour is 0.05. When we compute the loss landscape of one layer, the parameters of other layers are fixed. The gradient is computed based on 256 fixed samples since the gradient w.r.t. full dataset requires too much computation. Figure 3 and Figure 10 are centered at the final weight parameters, while others are centered at the initialization point to show the situation when training just starts. We visualize the loss landscapes on CUB-200, Stanford Cars and Food-101 for multiple times and reach consistent conclusions. But due to space limitation, we only show the results on one dataset for each experiment in the main paper. Other results are deferred to Section B.

**Computing the eigenvalue of Hessian.** We compute the eigenvalues of Hessian with Hessian-vector product and power methods based on the autograd of PyTorch. A similar implementation is provided by Yao et al. (2018). We only list top 20 eigenvalues in limited space.

**t-SNE embedding of model parameters.** We put the weight matrices of ResNet-50 in one vector as input. For faster computation, we pre-compute the distance between parameters of every two models with PyTorch and then use the distance matrix to compute the t-SNE embedding with scikit-learn. Note that we use the same ImageNet model from torchvision and the same Places model from Zhou et al. (2018) for fine-tuning.

**Variation of loss function in the direction of gradient.** Based on the original trajectory of training, we take steps in the direction of gradient from parameters at different steps during training to calculate the maximum changes of loss in that direction. The step size is set to the size of gradient. We take 100 steps from the original trajectories to measure the *local* property of loss landscapes. We aim to quantify the stability of loss functions and directly show the magnitude of gradient with this experiments on different datasets. Results on CUB-200 are provided in the main paper, with additional results further provided in Section B. Not that this experiment is inspired by Santurkar et al. (2018). We use the similar protocol as Section 3.2 in Santurkar et al. (2018). Another protocol is to fix the step size along the gradient and compute the maximum variation of loss. Results on Stanford Cars with this protocol are provided in Section B.2. Results for both scenarios are similar.

# B  ADDITIONAL EXPERIMENTAL RESULTS

## B.1  VISUALIZATION OF LANDSCAPES AROUND CONVERGENCE FOR DIFFERENT DATASETS

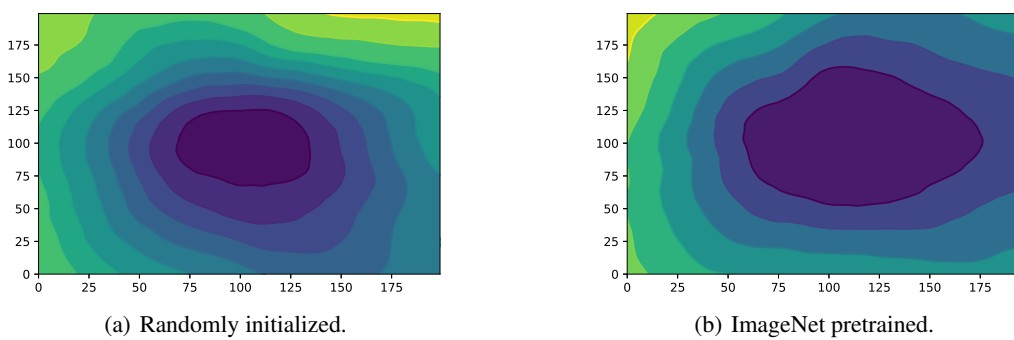

(a) Randomly initialized.

(b) ImageNet pretrained.

Figure 10: Comparison of landscapes centered at the minima on Food-101 datasets with ResNet-50. We use the filter normalization (Li et al., 2018a) to avoid the impact of weight scale. Randomly initialized networks end up with sharper minima, while pretrained networks stay in flat regions.

## B.2  VARIATION OF LOSS FUNCTION IN THE DIRECTION OF GRADIENT WITH FIXED STEP SIZE

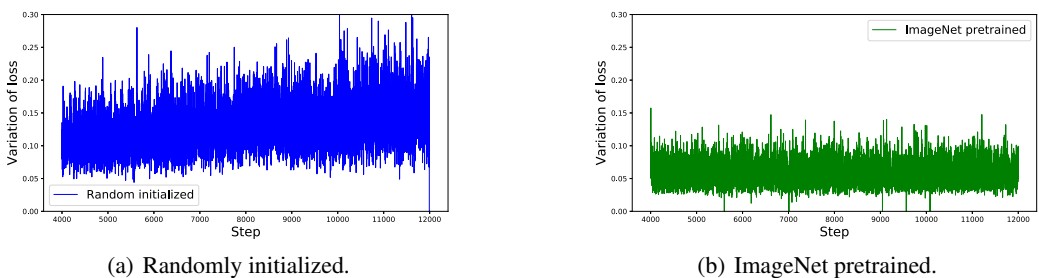

(a) Randomly initialized.

(b) ImageNet pretrained.

Figure 11: Variation of the loss in ResNet-50 with ImageNet pretrained weight and random initialization. We compare the variation of loss function in the direction of gradient during the training process on Stanford Cars dataset. The variation of pretrained networks is substantially smaller than the randomly initialized one, implying a more desirable loss landscape and more stable optimization.

## B.3  COMPARISON OF TRAINING FROM SCRATCH

To validate that the generalization error is indeed improved with pretraining, for each dataset, we list the generalization error and the norm of deviation from the pretrained parameters in Table 2. The decreased percentage is calculated by dividing the error reduced in fine-tuning with the error of training from scratch. Compared to the results of fine-tuning, we observe that ImageNet pretraining improves the generalization performance of general *coarse-grained* classification tasks significantly,

yet the performance boost is smaller for *fine-grained* tasks which are dissimilar in the sense of task with ImageNet. Note that, although Stanford Cars and CUB-200 are visually similar to ImageNet, what really matters is the similarity between the nature of tasks, i.e. both images and labels matter.

Table 2: Performance of training from scratch.

| Dataset | test error | fine-tuning improvement | $\frac{1}{\sqrt{n}} \sum_l \|\mathbf{W}_{(l)} - \mathbf{W}_{0(l)}\|_F$ |
|---|---|---|---|
| Webcam | 26.39±0.41 | 98.29% | 3.54±0.53 |
| Stanford Cars | 46.25±0.76 | 48.75% | 5.95±0.38 |
| Caltech-101 | 22.14±0.63 | 79.36% | 2.34±0.34 |
| CUB-200 | 52.70±0.77 | 60.15% | 3.90±0.61 |

### B.4 COMPARISON OF THE LOSS LANDSCAPES IN EACH LAYER

We visualize the loss landscape of 25-48th layers in ResNet-50 on Food-101 dataset. We compare the landscapes centered at the initialization point of randomly initialized and ImageNet pretrained networks – see Figure 12 and Figure 13. Results are in line with our observations of the magnitude of gradient in Figure 7. At higher layers, the landscapes of random initialization and ImageNet pretraining are similar. However, as the gradient is back-propagated through lower layers, the landscapes of pretrained networks remain as smooth as the higher layers. In sharp contrast, the landscapes of randomly initialized networks worsen through the lower layers, indicating that the magnitude of gradient is substantially worsened in back-propagation.

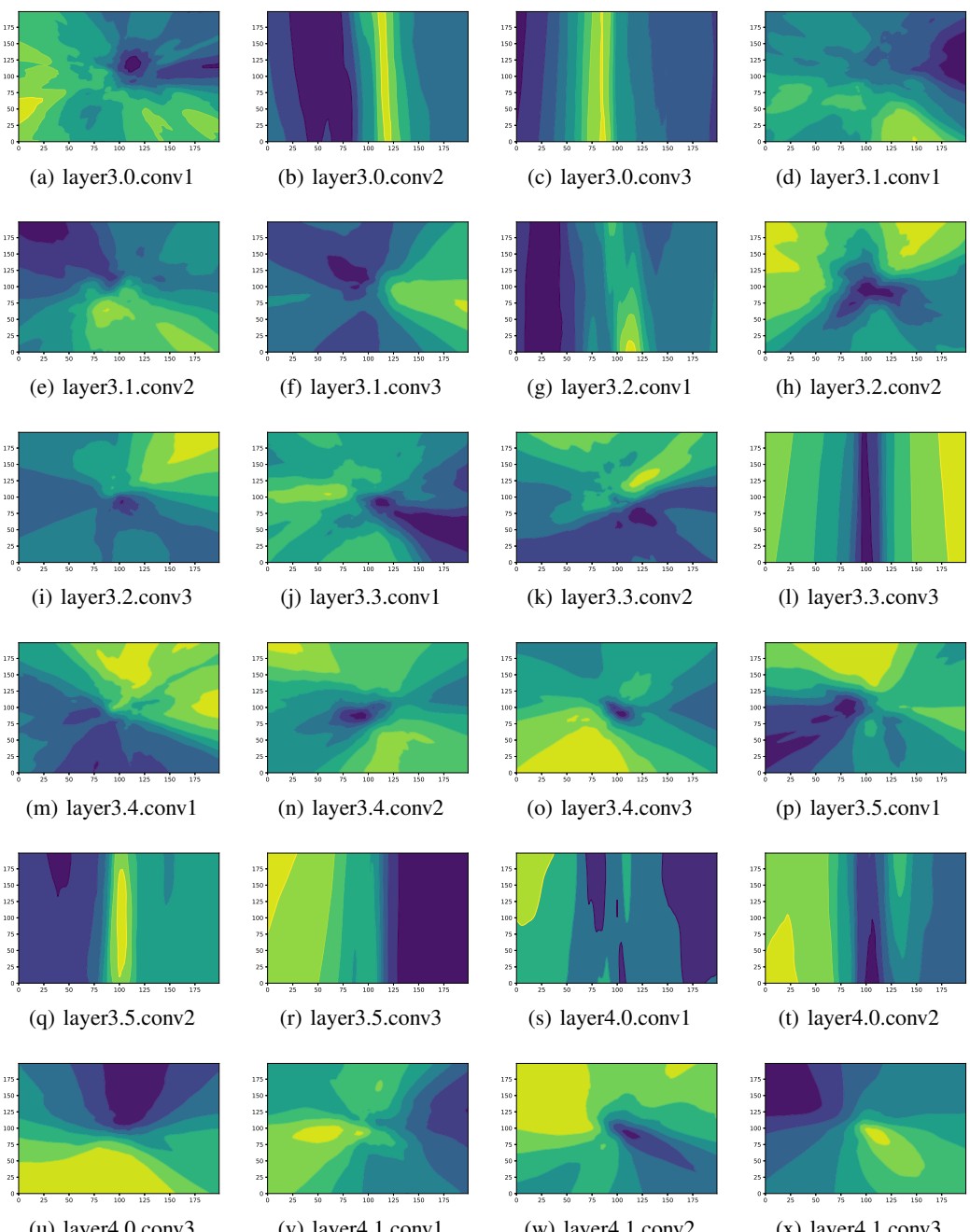

Figure 12: Landscapes centered at the initialization point of each layer in ResNet-50 using ImageNet pretrained weight. The smoothness of landscapes in each layer are nearly identical, indicating a proper scaling of gradient.

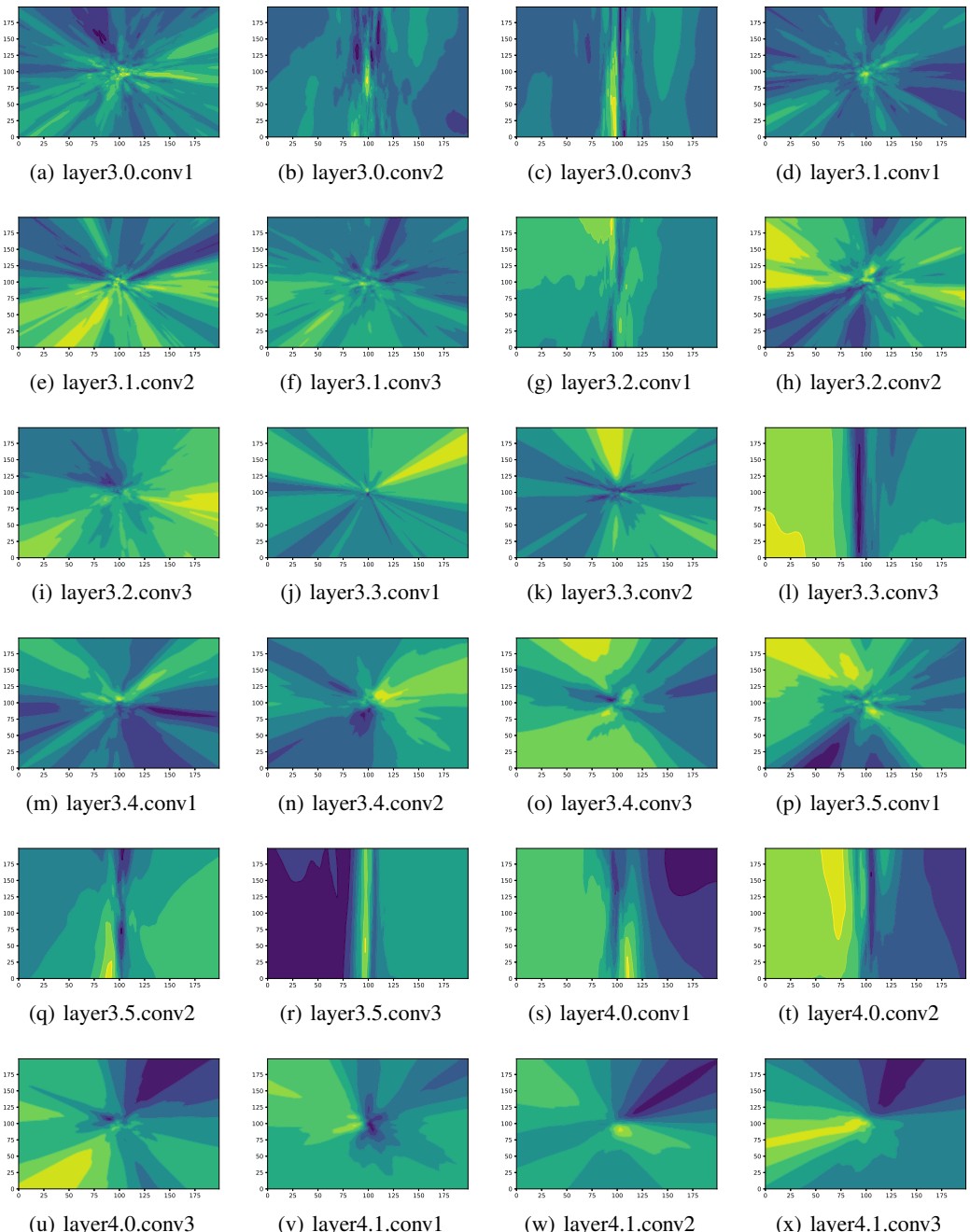

Figure 13: Landscapes centered at the initialization point of each layer in ResNet-50 initialized randomly. At the higher layers, the landscapes tend to be smooth. However, as the gradient is propagated to lower layers, the landscapes are becoming full of ridges and trenches in spite of the presence of Batch-Norm and skip connections.

## C PROOFS OF THEOREMS IN SECTION 6

To study how transferring pretrained knowledge helps target tasks, we first study the trajectories of weight matrices during pretraining and then analyze its effect as an initialization in target tasks. Our analysis is based on Du et al. (2019)'s framework for over-parametrized networks. For the weight matrix $\mathbf{W}$, $\mathbf{W}(0)$ denotes the random initialization. $\mathbf{W}_P(k)$ denotes $\mathbf{W}$ at the $k$th step of pretraining. $\mathbf{W}(P)$ denotes the pretrained weight matrix after training $K$ steps. $\mathbf{W}_Q(k)$ denotes the weight matrix after $K$ steps of fine-tuning from $\mathbf{W}(P)$. For other terms, the notation at each step is similar.

We first analyze the pretraining process on the source datasets based on Arora et al. (2019). Define a matrix $\mathbf{Z}_P \in \mathbb{R}^{md \times n_P}$ which is crucial to analyzing the trajectories of the weight matrix during pretraining,

$$\mathbf{Z}_P = \frac{1}{\sqrt{m}} \begin{pmatrix} \mathbb{I}_{1,1}^P a_1 \mathbf{x}_{P,1} & \cdots & \mathbb{I}_{1,n}^P a_1 \mathbf{x}_{P,n} \\ \vdots & \ddots & \vdots \\ \mathbb{I}_{m,1}^P a_m \mathbf{x}_{P,1} & \cdots & \mathbb{I}_{m,n}^P a_m \mathbf{x}_{P,n} \end{pmatrix} \in \mathbb{R}^{md \times n_P}, \tag{3}$$

where $\mathbb{I}_{i,j}^P = \mathbb{I}\{\mathbf{w}_i^\top \mathbf{x}_{P,j} \geq 0\}$. $\mathbf{Z}_P(k)$ denotes the matrix corresponding to $\mathbf{W}_P(k)$. Note that the gradient descent is carried out as

$$\mathrm{vec}(\mathbf{W}_P(k+1)) = \mathrm{vec}(\mathbf{W}_P(k)) - \eta \mathbf{Z}_P(k)(\mathbf{u}(k) - y_P), \tag{4}$$

where $\mathrm{vec}(\cdot)$ denotes concatenating a column of a matrice into a single vector. Then in the $K$ iterations of pretraining on the source dataset,

$$\mathrm{vec}(\mathbf{W}(P)) - \mathrm{vec}(\mathbf{W}(0))$$
$$= \sum_{k=0}^{K-1} \mathrm{vec}(\mathbf{W}_P(k+1)) - \mathrm{vec}(\mathbf{W}_P(k))$$
$$= -\eta \sum_{k=0}^{K-1} \mathbf{Z}_P(k)(\mathbf{u}_P(k) - \mathbf{y}_P)$$
$$= \sum_{k=0}^{K-1} \eta \mathbf{Z}_P(k)(\mathbf{I} - \eta \mathbf{H}_P^\infty)^k \mathbf{y}_P - \sum_{k=0}^{K-1} \eta \mathbf{Z}_P(k)\mathbf{e}_P(k)$$
$$= \sum_{k=0}^{K-1} \eta \mathbf{Z}_P(0)(\mathbf{I} - \eta \mathbf{H}_P^\infty)^k \mathbf{y} + \sum_{k=0}^{K-1} \eta(\mathbf{Z}_P(k) - \mathbf{Z}_P(0))(\mathbf{I} - \eta \mathbf{H}_P^\infty)^k \mathbf{y} - \sum_{k=0}^{K-1} \eta \mathbf{Z}_P(k)\mathbf{e}_P(k). \tag{5}$$

The first term is the primary component in the pretrained matrix, while the second and third terms is small under the over-parametrized conditions. Now following Arora et al. (2019), the magnitude of these terms can be bounded with probability no less than $1 - \delta$,

$$\|\boldsymbol{\epsilon}\|_2 = \|\sum_{k=0}^{K-1} \eta(\mathbf{Z}_P(k) - \mathbf{Z}_P(0))(\mathbf{I} - \eta \mathbf{H}^\infty)^k \mathbf{y} - \sum_{k=0}^{K-1} \eta \mathbf{Z}_P(k)\mathbf{e}_P(k)\|_2 = O\left(\frac{n_P \kappa}{\lambda_P \delta}\right) + O\left(\frac{n_P^2}{m^{\frac{1}{4}} \lambda_P^{\frac{3}{2}} \kappa^{\frac{1}{2}} \delta}\right). \tag{6}$$

Here we also provide lemmas from Du et al. (2019) which are extensively used later.

**Lemma 1.** *With $\lambda_P = \lambda_{\min}(\mathbf{H}_P^\infty) > 0$, $m = \Omega\left(\frac{n_P^6}{\lambda_P^4 \kappa^2 \delta^3}\right)$ and $\eta = O\left(\frac{\lambda_P}{n_P^2}\right)$, with probability at least $1 - \delta$ over the random initialization we have*

$$\|\mathbf{w}_{P,r}(k) - \mathbf{w}_r(0)\|_2 \leq \frac{4\sqrt{n_P}\|\mathbf{y}_P - \mathbf{u}_P(0)\|_2}{\sqrt{m}\lambda_P}, \quad \forall r \in [m], \forall k \geq 0. \tag{7}$$

**Lemma 2.** *If $\mathbf{w}_1, \ldots, \mathbf{w}_m$ are i.i.d. generated from $N(\mathbf{0}, \mathbf{I})$, then with probability at least $1 - \delta$, the following holds. For any set of weight vectors $\mathbf{w}_1, \ldots, \mathbf{w}_m \in \mathbb{R}^d$ that satisfy for any $r \in [m]$, $\|\mathbf{w}_r(0) - \mathbf{w}_r\|_2 \leq \frac{c\delta\lambda_0}{n^2} \triangleq R$ for some small positive constant $c$, then the matrix $\mathbf{H} \in \mathbb{R}^{n \times n}$ defined by*

$$\mathbf{H}_{ij} = \frac{1}{m} \mathbf{x}_i^\top \mathbf{x}_j \sum_{r=1}^m \mathbb{I}\{\mathbf{w}_r^\top \mathbf{x}_i \geq 0, \mathbf{w}_r^\top \mathbf{x}_j \geq 0\}$$

*satisfies $\|\mathbf{H} - \mathbf{H}(0)\|_2 < \frac{\lambda_0}{4}$ and $\lambda_{\min}(\mathbf{H}) > \frac{\lambda_0}{2}$.*

### C.1 PROOF OF THEOREM 1

Now we start to analyze the influence of pretrained weight on target tasks. 1) We show that during pretraining, $\mathbf{H}_{PQ}^{\infty}$ is close to $\mathbf{Z}_P(0)^{\top}\mathbf{Z}_Q(P)$. 2) Then we analyze $\mathbf{u}_Q(P) - \mathbf{u}_Q(0)$ with the properties of $\mathbf{H}_{PQ}^{\infty}$. 3) Standard calculation shows the magnitude of gradient relates closely to $\mathbf{u}_Q(P) - \mathbf{u}_Q(0)$, and we are able to find out how is the magnitude of gradient improved.

To start with, we analyze the properties of the matrix $\mathbf{H}_{PQ}^{\infty}$. We show that under over-parametrized conditions, $\mathbf{H}_{PQ}^{\infty}$ is close to the randomly initialized Gram matrix $\mathbf{Z}_P(0)^{\top}\mathbf{Z}_Q(P)$. Use $\mathbf{H}_{PQ}(0)$ to denote $\mathbf{Z}_P(0)^{\top}\mathbf{Z}_Q(0)$, and $\mathbf{H}_{PQ}(P)$ to denote $\mathbf{Z}_P(0)^{\top}\mathbf{Z}_Q(P)$.

**Lemma 3.** *With the same condition as lemma 1, with probability no less than $1 - \delta$,*

$$\|\mathbf{H}_{PQ}(P) - \mathbf{H}_{PQ}^{\infty}\|_F \leq O\left(\frac{n_P^2 n_Q}{\sqrt{m}\lambda_P\kappa\delta^{3/2}}\right). \tag{8}$$

$$|\mathbf{H}_{PQ,ij}(P) - \mathbf{H}_{PQ,ij}(0)| = \left|\frac{\mathbf{x}_{P,i}^{\top}\mathbf{x}_{Q,j}}{m}\sum_{r=1}^{m}(\mathbb{I}_{ri}^P(0)\mathbb{I}_{rj}^Q(P) - \mathbb{I}_{ri}^P(0)\mathbb{I}_{ri}^Q(0))\right|$$

$$\leq \frac{1}{m}\sum_{r=1}^{m}\mathbb{I}(\mathbb{I}_{ri}^Q(P) \neq \mathbb{I}_{ri}^Q(0)) \tag{9}$$

$$\leq \frac{1}{m}\sum_{r=1}^{m}(\mathbb{I}(|\mathbf{w}_r(0)^{\top}\mathbf{x}_{Q,i}| \leq R) + \mathbb{I}(\|\mathbf{w}_r(P) - \mathbf{w}_r(0)\|_2 > R)),$$

where $R = \frac{c\sqrt{n_P}\|\mathbf{y}_P - \mathbf{u}_P(0)\|_2}{\sqrt{m}\lambda_P}$ with a small $c$. Since $\mathbf{w}_r(0)$ is independent of $\mathbf{x}_{Q,i}$ and $\|\mathbf{x}_{Q,i}\|_2 = 1$, the distribution of $\mathbf{w}(0)_r^{\top}\mathbf{x}_{Q,i}$ and $\mathbf{w}_r(0)$ are the same Gaussian. $\mathbb{E}[\mathbb{I}(|\mathbf{w}_r(0)^{\top}\mathbf{x}_{Q,i}| \leq R)] = \mathbb{E}[\mathbb{I}(|\mathbf{w}_r(0)| \leq R)] \leq \frac{2R}{\sqrt{2\pi\kappa}}$.

$$\mathbb{E}[|\mathbf{H}_{PQ,ij}(P) - \mathbf{H}_{PQ,ij}(0)|] \leq \frac{2R}{\sqrt{2\pi\kappa}} + \frac{\delta}{m}. \tag{10}$$

Applying Markov's inequality, and noting that $\|\mathbf{y}_P - \mathbf{u}(0)\|_2 = O\left(\frac{n_P}{\delta}\right)$ we have with probability no less than $1 - \delta$,

$$\|\mathbf{H}_{PQ}(P) - \mathbf{H}_{PQ}(0)\|_F \leq \frac{n_P n_Q}{\delta}\left(\frac{2R}{\sqrt{2\pi\kappa}} + \frac{\delta}{m}\right) = O\left(\frac{n_P^2 n_Q}{\sqrt{m}\lambda_P\kappa\delta^{3/2}}\right). \tag{11}$$

Also note that $\mathbb{E}[\mathbf{H}_{PQ,ij}(0)] = \mathbf{H}_{PQ,ij}^{\infty}$. By Hoeffding's inequality, we have with probability at least $1 - \delta$,

$$\|\mathbf{H}_{PQ}^{\infty} - \mathbf{H}_{PQ}(0)\|_F \leq \frac{n_P n_Q \log(2n_P n_Q/\delta)}{2m}. \tag{12}$$

Combining equation 12 and equation 11, we have with probability at least $1 - \delta$,

$$\|\mathbf{H}_{PQ}(P) - \mathbf{H}_{PQ}^{\infty}\|_F \leq O\left(\frac{n_P^2 n_Q}{\sqrt{m}\lambda_P\kappa\delta^{3/2}}\right).$$

$\square$

Denote by $\mathbf{u}_Q(P), \mathbf{u}_Q(0)$ the output on the target dataset using weight matrix $\mathbf{W}(P)$ and $\mathbf{W}_0$ respectively. First, we compute the gradient with respect to the activations,

$$\frac{\partial L(\mathbf{W}(P))}{\partial \mathbf{X}^1} = \frac{1}{\sqrt{m}}\mathbf{a}(\mathbf{u}_Q(P) - \mathbf{y}_Q), \tag{13}$$

$$\|\frac{\partial L(\mathbf{W}(P))}{\partial \mathbf{X}^1}\|_2^2 = \frac{1}{m}\mathbf{a}^{\top}\mathbf{a}(\mathbf{u}_Q(P) - \mathbf{y}_Q)^2$$

$$= \|\mathbf{u}_Q(0) - \mathbf{y}_Q\|_2^2 + \|\mathbf{u}_Q(P) - \mathbf{u}_Q(0)\|_2^2 + 2\langle\mathbf{u}_Q(P) - \mathbf{u}_Q(0), \mathbf{u}_Q(0) - \mathbf{y}_Q\rangle. \tag{14}$$

It is obvious from equation 14 that $\mathbf{u}_Q(P) - \mathbf{u}_Q(0)$ should become the focus of our analysis. To calculate $\mathbf{u}_Q(P) - \mathbf{u}_Q(0)$, we need to sort out how the activations change by initializing the target networks with $\mathbf{W}(P)$ instead of $\mathbf{W}(0)$.

$$
\begin{aligned}
\mathbf{u}_Q(P) - \mathbf{u}_Q(0) &= \frac{1}{\sqrt{m}}(\mathbf{a}^\top(\sigma(\mathbf{W}(P)^\top\mathbf{X}) - \sigma(\mathbf{W}^\top(0)\mathbf{X})))^\top \\
&= \frac{1}{\sqrt{m}}\sum_{r=1}^m a_r(\sigma(\mathbf{w}_{P,r}^\top\mathbf{X}_Q) - \sigma(\mathbf{w}_r^\top(0)\mathbf{X}_Q))
\end{aligned}
\tag{15}
$$

For each $\mathbf{x}_{Q,i}$, divide $r$ into two sets to quantify the change of variation in activations on the target dataset.

$$
S_i = \{r \in [m], |\mathbf{w}_r(0)^\top\mathbf{x}_{Q,i}| \geq R\}, \overline{S}_i = \{r \in [m], |\mathbf{w}(0)_r^\top\mathbf{x}_{Q,i}| \leq R\},
\tag{16}
$$

where $R = \frac{4\sqrt{n}_P\|\mathbf{y}_P - \mathbf{u}(0)\|_2}{\sqrt{m}\lambda_P}$. For $r$ in $\overline{S}_i$, we can estimate the size of $\overline{S}_i$. Note that $\mathbb{E}[\overline{S}_i] = \mathbb{E}\left[\sum_{i=1}^n\sum_{r=1}^m \mathbb{I}(|\mathbf{w}(0)_r^\top\mathbf{x}_{Q,i}| \leq R)\right]$. For each $i$ and $r$, $\mathbb{E}[\mathbb{I}(|\mathbf{w}(0)_r^\top\mathbf{x}_{Q,i}| \leq R)] = \mathbb{E}[\mathbb{I}(|\mathbf{w}(0)_r| \leq R)] \leq \frac{2R}{\sqrt{2\pi}\kappa}$, since the distribution of $\mathbf{w}(0)_r$ is Gaussian with mean 0 and covariance matrix $\kappa^2\mathbf{I}$. Therefore, taking sum over all $i$ and $m$ and using Markov inequality, with probability at least $1 - \delta$ over the random initialization we have

$$
|\overline{S}_i| \leq \frac{2mn_PR}{\sqrt{2\pi}\kappa\delta} = \frac{8\sqrt{mn_P}\|\mathbf{y}_P - \mathbf{u}_P(0)\|_2}{\sqrt{2\pi}\kappa\lambda_P\delta}, \quad \forall r \in [m], \forall k \geq 0.
\tag{17}
$$

Thus, this part of activations is the same for $\mathbf{W}(0)$ and $\mathbf{W}(P)$ on the target dataset. For each $\mathbf{x}_{Q,i}$,

$$
\begin{aligned}
u_{Q,i}(P) - u_{Q,i}(0) &= \frac{1}{\sqrt{m}}\sum_{r=1}^m a_r(\sigma(\mathbf{w}_{P,r}^\top\mathbf{x}_{Q,i}) - \sigma(\mathbf{w}_r^\top(0)\mathbf{x}_{Q,i})) \\
&= \frac{1}{\sqrt{m}}\sum_{r\in[m]} a_r(\mathbb{I}_{r,i}^Q(0)(\mathbf{w}_{P,r}^\top\mathbf{x}_{Q,i}) - \mathbb{I}_{r,i}^Q(0)(\mathbf{w}_r^\top(0)\mathbf{x}_{Q,i})) \\
&\quad + \frac{1}{\sqrt{m}}\sum_{r\in\overline{S}_i} a_r(\sigma(\mathbf{w}_{P,r}^\top\mathbf{x}_{Q,i}) - \sigma(\mathbf{w}_r^\top(0)\mathbf{x}_{Q,i})) \\
&\quad - \frac{1}{\sqrt{m}}\sum_{r\in\overline{S}_i} a_r(\mathbb{I}_{r,i}^Q(0)(\mathbf{w}_{P,r}^\top\mathbf{x}_{Q,i}) - \mathbb{I}_{r,i}^Q(0)(\mathbf{w}_r^\top(0)\mathbf{x}_{Q,i})),
\end{aligned}
\tag{18}
$$

where $\mathbb{I}_{r,i}^Q(0)$ denotes $\mathbb{I}\{\mathbf{w}_r^\top(0)\mathbf{x}_{Q,i} \geq 0\}$. The first term is the primary part, while we can show that the second and the third term can be bounded with $\frac{1}{\sqrt{m}}|\overline{S}_i|\|\mathbf{w}_r(P) - \mathbf{w}_r(0)\|_2$ since $\|\mathbf{x}_{Q,i}\|_2 = 1$. Putting all $\mathbf{x}_{Q,i}$ together,

$$
\begin{aligned}
\mathbf{u}_Q(P) &- \mathbf{u}_Q(0) \\
&= \frac{1}{\sqrt{m}}(\mathbf{a}^\top\sigma_0((\mathbf{W}(P) - \mathbf{W}(0))^\top\mathbf{X}) + \boldsymbol{\epsilon}_1 + \boldsymbol{\epsilon}_2)^\top \\
&= \mathbf{Z}_Q(0)^\top \text{vec}(\mathbf{W}(P) - \mathbf{W}(0)) + \boldsymbol{\epsilon}_1 + \boldsymbol{\epsilon}_2 \\
&= \mathbf{Z}_Q(0)^\top(\mathbf{Z}_P(0)(\mathbf{H}_P^\infty)^{-1}\mathbf{y}_P + \boldsymbol{\epsilon}) + \boldsymbol{\epsilon}_1 + \boldsymbol{\epsilon}_2,
\end{aligned}
\tag{19}
$$

where $\boldsymbol{\epsilon}_1$ and $\boldsymbol{\epsilon}_2$ correspond to each of the second term and third term in equation 18. Thus, using lemma 1 and the estimation of $|\overline{S}_i|$, with probability no less than $1 - \delta$,

$$
\|\boldsymbol{\epsilon}'\|_2 = \|\boldsymbol{\epsilon}_1 + \boldsymbol{\epsilon}_2\|_2 \leq \frac{\sqrt{n_Q}}{\sqrt{m}}|\overline{S}_i|\|\mathbf{w}_r(P) - \mathbf{w}_r(0)\|_2 = O\left(\frac{n_P^2n_Q^{1/2}}{\sqrt{m}\lambda_P^2\delta^2\kappa}\right).
\tag{20}
$$

Now equipped with equation 6, equation 19, equation 20 and lemma 3, we are ready to calculate exactly how much pretrained wight matrix $\mathbf{W}(P)$ help reduce the magnitude of gradient over $\mathbf{W}(0)$,

$$
\begin{aligned}
\|\frac{\partial L}{\partial \mathbf{X}^1}\|_2^2 &= \frac{1}{m}\mathbf{a}^\top \mathbf{a}(\mathbf{u}_Q(P) - \mathbf{y}_Q)^2 \\
&= \|\mathbf{u}_Q(0) - \mathbf{y}_Q\|_2^2 + \|\mathbf{u}_Q(P) - \mathbf{u}_Q(0)\|_2^2 + 2\langle \mathbf{u}_Q(P) - \mathbf{u}_Q(0), \mathbf{u}_Q(0) - \mathbf{y}_Q \rangle \\
&= \|\frac{\partial L_0}{\partial \mathbf{X}^1}\|_2^2 + \mathbf{y}_p^\top \mathbf{H}_P^{\infty-1} \mathbf{H}_{PQ}^\infty \mathbf{H}_{PQ}^{\infty\top} \mathbf{H}_P^{\infty-1}\mathbf{y}_P - 2\mathbf{y}_Q^\top \mathbf{H}_{PQ}^{\infty\top} \mathbf{H}_P^{\infty-1}\mathbf{y}_P \\
&\quad + \|\boldsymbol{\epsilon}'\|_2^2 + 2\boldsymbol{\epsilon}'^\top \mathbf{Z}_Q(P)^\top \mathbf{Z}_P(0)\mathbf{H}_P^{\infty-1}\mathbf{y}_P + 2\boldsymbol{\epsilon}'^\top \mathbf{Z}_Q(P)^\top \boldsymbol{\epsilon} \\
&\quad + \|\mathbf{y}_P^\top \mathbf{H}_P^{\infty-1}(\mathbf{Z}_P(0)^\top \mathbf{Z}_Q(P) - \mathbf{H}_{PQ}^\infty)\|_2^2 + \|\boldsymbol{\epsilon}\|_2^2 + 2\boldsymbol{\epsilon}^\top \mathbf{Z}_Q(P)^\top \mathbf{Z}_P(0)\mathbf{H}_P^{\infty-1}\mathbf{y}_P \\
&\quad + \mathbf{u}_Q(0)^\top \mathbf{Z}_Q(P)^\top \mathbf{Z}_P(0)\mathbf{H}_P^{\infty-1}\mathbf{y}_P + \mathbf{u}_Q(0)^\top \mathbf{Z}_Q(P)^\top \boldsymbol{\epsilon} + \mathbf{u}_Q(0)^\top \boldsymbol{\epsilon}' \\
&\quad + 2\mathbf{y}_Q(\mathbf{Z}_Q(P)^\top \mathbf{Z}_P(0) - \mathbf{H}_{PQ}^{\infty\top})\mathbf{H}_P^{\infty-1}\mathbf{y}_P.
\end{aligned}
\tag{21}
$$

In equation 21, note that $\|\boldsymbol{\epsilon}\|_2$, $\|\boldsymbol{\epsilon}'\|_2$, $\|\mathbf{Z}_Q(P)^\top \mathbf{Z}_P(0) - \mathbf{H}_{PQ}^{\infty\top}\|_F = \|\mathbf{H}_{PQ}(P) - \mathbf{H}_{PQ}^\infty\|_F$, and $\|\mathbf{u}_Q(0)\|_2$ are all small values we have estimated above. Therefore, using $\|\mathbf{Z}_P(0)\|_F \leq \sqrt{n_P}$ and $\|\mathbf{Z}_Q(P)\|_F \leq \sqrt{n_Q}$, we can control the magnitude of the perturbation terms under over-parametrized conditions. Concretely, with probability at least $1 - \delta$ over random initialization,

$$
\|\boldsymbol{\epsilon}'\|_2^2 = O\left(\frac{n_P^4 n_Q}{m\lambda_P^4 \delta^4 \kappa^2}\right)
\tag{22}
$$

$$
\boldsymbol{\epsilon}'^\top \mathbf{Z}_Q(P)^\top \mathbf{Z}_P(0)\mathbf{H}_P^{\infty-1}\mathbf{y}_P = O\left(\frac{n_P^2 n_Q^{1/2}}{\sqrt{m}\lambda_P^2 \delta\kappa}\sqrt{n_P}\sqrt{n_Q}\frac{1}{\lambda_P}\sqrt{n_P}\right) = O\left(\frac{n_P^3 n_Q}{\sqrt{m}\lambda_P^3 \delta^2 \kappa}\right)
\tag{23}
$$

$$
\boldsymbol{\epsilon}'^\top \mathbf{Z}_Q(P)^\top \boldsymbol{\epsilon} = O\left(\frac{n_P^3 n_Q}{\sqrt{m}\lambda_P^3 \delta^3}\right) + O\left(\frac{n_P^3 n_Q}{m^{3/4}\lambda_P^{7/2}\delta^3 \kappa^{3/2}}\right)
\tag{24}
$$

$$
\|\mathbf{y}_P^\top \mathbf{H}_P^{\infty-1}(\mathbf{Z}_P(0)^\top \mathbf{Z}_Q(P) - \mathbf{H}_{PQ}^\infty)\|_2^2 = O\left(\sqrt{n_P}\frac{1}{\lambda_P}\frac{n_P^2 n_Q}{\sqrt{m}\lambda_P \kappa \delta^{3/2}}\right)^2 = O\left(\frac{n_P^5 n_Q^2}{m\lambda_P^4 \kappa^2 \delta^3}\right)
\tag{25}
$$

$$
\|\boldsymbol{\epsilon}\|_2^2 = O\left(\frac{n_P^2 \kappa^2}{\lambda_P^2 \delta^2}\right) + O\left(\frac{n_P^4}{m^{1/2}\lambda_P^3 \kappa\delta^2}\right)
\tag{26}
$$

$$
\boldsymbol{\epsilon}^\top \mathbf{Z}_Q(P)^\top \mathbf{Z}_P(0)\mathbf{H}_P^{\infty-1}\mathbf{y}_P = O\left(\frac{n_P^2 n_Q^{1/2}\kappa}{\lambda_P^2 \delta}\right) + O\left(\frac{n_P^3 n_Q^{1/2}}{m^{1/4}\lambda_P^{5/2}\sqrt{\kappa}\delta}\right)
\tag{27}
$$

$$
\mathbf{u}_Q(0)^\top \mathbf{Z}_Q(P)^\top \mathbf{Z}_P(0)\mathbf{H}_P^{\infty-1}\mathbf{y}_P = O\left(\frac{n_P n_Q \kappa}{\lambda_P \sqrt{\delta}}\right)
\tag{28}
$$

$$
\mathbf{u}_Q(0)^\top \mathbf{Z}_Q(P)^\top \boldsymbol{\epsilon} = O\left(\frac{n_P n_Q \kappa^2}{\lambda_P \delta^{3/2}}\right) + O\left(\frac{n_P^2 n_Q}{m^{1/4}\lambda_P^{3/2}\sqrt{\kappa}\delta^{3/2}}\right)
\tag{29}
$$

$$
\mathbf{u}_Q(0)^\top \boldsymbol{\epsilon}' = O\left(\frac{n_P^2 n_Q}{m^{1/2}\lambda_P^2 \delta^{5/2}}\right)
\tag{30}
$$

$$
\mathbf{y}_Q(\mathbf{Z}_Q(P)^\top \mathbf{Z}_P(0) - \mathbf{H}_{PQ}^{\infty\top})\mathbf{H}_P^{\infty-1}\mathbf{y}_P = O\left(\frac{n_P^{5/2} n_Q^{3/2}}{\sqrt{m}\lambda_P^2 \kappa\delta^{3/2}}\right)
\tag{31}
$$

Substituting these estimations into equation 21 completes the proof of Theorem 1. $\qquad\square$

## C.2  PROOF OF THEOREM 2

In this subsection, we analyze the impact of pretrained weight matrix on the generalization performance. First, we show that a model will converge if initialized with pretrained weight matrix. Based on this, we further investigate the trajectories during transfer learning and bound $\|\mathbf{W} - \mathbf{W}(P)\|_F$ with the relationship between source and target datasets.

C.2.1 CONVERGENCE OF TRANSFERRING FROM PRETRAINED REPRESENTATIONS

Similar to Du et al. (2019), the proof is done with induction, but since we start from $\mathbf{W}(P)$ instead of randomly initialized $\mathbf{W}_0$ in the transferring process, we should use the randomness of $\mathbf{W}_0$ in an indirect way. Concretely, we should use lemma 1 to bound the difference between each column of $\mathbf{W}(P)$ and randomly initialized $\mathbf{W}(0)$ when proving the induction hypothesis.

**Theorem 3** (Convergence of Transfer Learning). *Under the same conditions as in Theorem 1, if we set the number of hidden nodes* $m = \Omega\left(\frac{n_P^8 n_Q^6}{\lambda_P^{16}\lambda_P^4\kappa^2\delta^{10}}\right)$, $\kappa = O\left(\frac{\lambda_P^2\delta}{n_P^2 n_Q^{\frac{1}{2}}}\right)$, *and the learning rate* $\eta = O\left(\frac{\lambda_Q}{n_Q^2}\right)$ *then with probability at least* $1 - \delta$ *over the random initialization we have for* $k = 0, 1, 2, \ldots$

$$\|\mathbf{u}_Q(k) - \mathbf{y}_Q\|_2^2 \leq \left(1 - \frac{\eta\lambda_Q}{2}\right)^k \|\mathbf{u}_Q(P) - \mathbf{y}_Q\|_2^2. \tag{32}$$

The following lemma is a direct corollary of Theorem 3 and lamma 1, and is crucial to analysis to follow.

**Lemma 4.** *Under the same conditions as Theorem 3, with probability at least* $1 - \delta$ *over the random initialization we have* $\forall r \in [m], \forall k \geq 0$,

$$\|\mathbf{w}_{Q,r}(k) - \mathbf{w}_r(0)\|_2 \leq \frac{4\sqrt{n_P}\|\mathbf{y}_P - \mathbf{u}_P(0)\|_2}{\sqrt{m}\lambda_P} + \frac{4\sqrt{n_Q}\|\mathbf{y}_Q - \mathbf{u}_Q(P)\|_2}{\sqrt{m}\lambda_Q} = O\left(\frac{n_P^3 n_Q^{\frac{3}{2}}}{\sqrt{m}\lambda_P^2\lambda_Q\delta^{\frac{3}{2}}}\right). \tag{33}$$

We have the estimation of $\|\mathbf{w}_{Q,r}(k) - \mathbf{w}_r(0)\|_2$ from lemma 1. From $\|\mathbf{w}_{Q,r}(k) - \mathbf{w}_r(0)\|_2 \leq \|\mathbf{w}_{Q,r}(k) - \mathbf{w}_r(P)\|_2 + \|\mathbf{w}_r(P) - \mathbf{w}_r(0)\|_2$, we can proove lemma 4 by estimating $\|\mathbf{w}_{Q,r}(k) - \mathbf{w}_r(P)\|_2$.

$$\begin{aligned}
\|\mathbf{w}_{Q,r}(k) - \mathbf{w}_r(P)\|_2 &= \eta \sum_{k'=0}^{k} \|\frac{\partial L(\mathbf{W}(k'))}{\partial \mathbf{w}_r(k')}\|_2 \\
&\leq \eta \sum_{k'=0}^{k} \frac{\sqrt{n_Q}\|\mathbf{y}_Q - \mathbf{u}_Q(k')\|_2}{\sqrt{m}} \\
&\leq \eta \sum_{k'=0}^{\infty} \frac{\sqrt{n_Q}(1 - \frac{\eta\lambda_Q}{2})^{k'/2}}{\sqrt{m}}\|\mathbf{y}_Q - \mathbf{u}_Q(k')\|_2 \\
&= \frac{4\sqrt{n_Q}\|\mathbf{y}_Q - \mathbf{u}_Q(P)\|_2}{\sqrt{m}\lambda_Q}
\end{aligned}$$

We also have $\|\mathbf{y}_Q - \mathbf{u}_Q(0)\|_2 = O\left(\kappa\frac{\sqrt{n_Q}}{\sqrt{\delta}}\right)$, and $\mathbf{u}_Q(P) - \mathbf{u}_Q(0) = \mathbf{Z}_Q(0)^\top(\mathbf{Z}_P(0)\mathbf{H}_P^{\infty-1}\mathbf{y}_P + \epsilon) + \epsilon_1 + \epsilon_2$. Substituting lemma 3, equation 6, and equation 22 into $\|\mathbf{u}_Q(P) - \mathbf{u}_Q(0)\|_2$ completes the proof. $\square$

Now we start to prove Theorem 3 by induction.

**Condition 1.** *At the $k$-th iteration, we have* $\|\mathbf{u}_Q(k) - \mathbf{y}_Q\|_2^2 \leq \left(1 - \frac{\eta\lambda_{0Q}}{2}\right)^k \|\mathbf{u}_Q(P) - \mathbf{y}_Q\|_2^2$.

We have the following corollary if condition 1 holds,

**Corollary 1.** *If condition 1 holds for* $k' = 0, \ldots, k$, *for every* $r \in [m]$, *with probability at least* $1 - \delta$,

$$\|\mathbf{w}_{Q,r}(k) - \mathbf{w}_r(0)\|_2 \leq \frac{4\sqrt{n_P}\|\mathbf{y}_P - \mathbf{u}_P(0)\|_2}{\sqrt{m}\lambda_P} + \frac{4\sqrt{n_Q}\|\mathbf{y}_Q - \mathbf{u}_Q(P)\|_2}{\sqrt{m}\lambda_Q} \triangleq R'.$$

If $k = 0$, by definition Condition 1 holds. Suppose for $k' = 0, \ldots, k$, condition 1 holds and we want to show it still holds for $k' = k + 1$. The strategy is similar to the proof of convergence

on training from scratch. By classifying the change of activations into two categories, we are able to deal with the ReLU networks as a perturbed version of linear regression. We define the event $A_{ir} = |\mathbf{w}_r(0)^\top \mathbf{x}_{Q,i} \le R|$, where $R = \frac{c\lambda_Q}{n_Q^2}$ for some small positive constant $c$ to control the magnitude of perturbation. Similar to the analysis above, we let $S_i = \{r \in [m] : \mathbb{I}\{A_{ir}\} = 0\}$ and $\overline{S}_i = [m] \setminus S_i$. Since the distribution of $\mathbf{w}_r(0)$ is Gaussian, we can bound the value of each $A_{ir}$ and then bound the size of $\overline{S}_i$ just as we have estimated in equation 17 above.

**Lemma 5.** *With probability at least $1 - \delta$ over the initialization, we have $\sum_{i=1}^n |\overline{S}_i| \le \frac{Cmn_Q R}{\delta}$ for some positive constant $C > 0$.*

The following analysis is identical to the situation of training from scratch.

$$u_{Q,i}(k+1) - u_{Q,i}(k) = \frac{1}{\sqrt{m}} \sum_{r=1}^m a_r \left( \sigma \left( \mathbf{w}_{Q,r}(k+1)^\top \mathbf{x}_{Q,i} \right) - \sigma \left( \mathbf{w}_{Q,r}(k)^\top \mathbf{x}_{Q,i} \right) \right).$$

By dividing $[m]$ into $S_i$ and $\overline{S}_i$, we have,

$$I_1^i \triangleq \frac{1}{\sqrt{m}} \sum_{r \in S_i} a_r \left( \sigma \left( \mathbf{w}_{Q,r}(k+1)^\top \mathbf{x}_{Q,i} \right) - \sigma \left( \mathbf{w}_{Q,r}(k)^\top \mathbf{x}_{Q,i} \right) \right)$$

$$I_2^i \triangleq \frac{1}{\sqrt{m}} \sum_{r \in \overline{S}_i} a_r \left( \sigma \left( \mathbf{w}_{Q,r}(k+1)^\top \mathbf{x}_{Q,i} \right) - \sigma \left( \mathbf{w}_{Q,r}(k)^\top \mathbf{x}_{Q,i} \right) \right).$$

We view $I_2^i$ as a perturbation and bound its magnitude. Because ReLU is a 1-Lipschitz function and $|a_r| = 1$, we have

$$|I_2^i| \le \frac{\eta}{m^{\frac{1}{2}}} \sum_{r \in \overline{S}_i} |\left( \frac{\partial L(\mathbf{W}_Q(k))}{\partial \mathbf{w}_{Q,r}(k)} \right)^\top \mathbf{x}_{Q,i}| \le \frac{\eta |\overline{S}_i|}{\sqrt{m}} \max_{r \in [m]} \| \frac{\partial L(\mathbf{W}_Q(k))}{\partial \mathbf{w}_{Q,r}(k)} \|_2 \le \frac{\eta |\overline{S}_i| n_Q^{\frac{1}{2}} \| \mathbf{u}_Q(k) - \mathbf{y}_Q \|_2}{m}.$$

By Corollary 1, we know $\| \mathbf{w}_{Q,r}(k) - \mathbf{w}_r(0) \| \le R'$ for all $r \in [m]$. Furthermore, with the conditions on $m$, we have $R' < R$. Thus $\mathbb{I}\{ \mathbf{w}_{Q,r}(k+1)^\top \mathbf{x}_{Q,i} \ge 0 \} = \mathbb{I}\{ \mathbf{w}_{Q,r}(k)^\top \mathbf{x}_{Q,i} \ge 0 \}$ for $r \in S_i$..

$$I_1^i = -\frac{\eta}{m} \sum_{j=1}^{n_Q} \mathbf{x}_{Q,i}^\top \mathbf{x}_{Q,j} (u_{Q,j} - y_{Q,j}) \sum_{r \in S_i} \mathbb{I}\{ \mathbf{w}_{Q,r}(k)^\top \mathbf{x}_{Q,i} \ge 0, \mathbf{w}_{Q,r}(k)^\top \mathbf{x}_{Q,j} \ge 0 \}$$

$$= -\frac{\eta}{m} \sum_{j=1}^{n_Q} \mathbf{x}_{Q,i}^\top \mathbf{x}_{Q,j} (u_{Q,j} - y_{Q,j}) \sum_{r=1}^m \mathbb{I}\{ \mathbf{w}_{Q,r}(k)^\top \mathbf{x}_{Q,i} \ge 0, \mathbf{w}_{Q,r}(k)^\top \mathbf{x}_{Q,j} \ge 0 \}$$

$$+ \frac{\eta}{m} \sum_{j=1}^{n_Q} \mathbf{x}_{Q,i}^\top \mathbf{x}_{Q,j} (u_{Q,j} - y_{Q,j}) \sum_{r \in \overline{S}_i} \mathbb{I}\{ \mathbf{w}_{Q,r}(k)^\top \mathbf{x}_{Q,i} \ge 0, \mathbf{w}_{Q,r}(k)^\top \mathbf{x}_{Q,j} \ge 0 \}$$

where $\frac{1}{m} \sum_{r=1}^m \mathbf{x}_{Q,i}^\top \mathbf{x}_{Q,j} \mathbb{I}\{ \mathbf{w}_{Q,r}(k)^\top \mathbf{x}_{Q,i} \ge 0, \mathbf{w}_{Q,r}(k)^\top \mathbf{x}_{Q,j} \ge 0 \}$ is just the $(i,j)$-th entry of a discrete version of Gram matrix $\mathbf{H}_Q^\infty$ defined in Section 6.1 and

$$| \frac{\eta}{m} \sum_{j=1}^{n_Q} \mathbf{x}_{Q,i}^\top \mathbf{x}_{Q,j} (u_{Q,j} - y_{Q,j}) \sum_{r \in \overline{S}_i} \mathbb{I}\{ \mathbf{w}_{Q,r}(k)^\top \mathbf{x}_{Q,i} \ge 0, \mathbf{w}_{Q,r}(k)^\top \mathbf{x}_{Q,j} \ge 0 \} |$$

$$\le \frac{\eta}{m} |\overline{S}_i| \sum_{j=1}^{n_Q} |u_{Q,j}(k) - y_{Q,j}| \le \frac{\eta \sqrt{n_Q}}{m} |\overline{S}_i| \| \mathbf{u}_Q(k) - \mathbf{y}_Q \|_2.$$

For ease of notations, denote by $\| \mathbf{H}_Q(k)^\perp \|_2$ the matrix whose $i, j$ entry is $\frac{1}{m} \mathbf{x}_{Q,i}^\top \mathbf{x}_{Q,j} (u_{Q,j} - y_{Q,j}) \sum_{r \in \overline{S}_i} \mathbb{I}\{ \mathbf{w}_{Q,r}(k)^\top \mathbf{x}_{Q,i} \ge 0, \mathbf{w}_{Q,r}(k)^\top \mathbf{x}_{Q,j} \ge 0 \}$. Therefore, the $L2$ norm of $\| \mathbf{H}(k)^\perp \|_2$ is bounded with $\frac{Cn_Q^2 R}{\delta}$. To bound the quadratic term, we use the same techniques as training from scratch.

$$\| \mathbf{u}_Q(k+1) - \mathbf{u}_Q(k) \|_2^2 \le \eta^2 \sum_{i=1}^{n_Q} \frac{1}{m} \left( \sum_{r=1}^m \| \frac{\partial L(\mathbf{W}_Q(k))}{\partial \mathbf{w}_{Q,r}(k)} \|_2 \right)^2 \le \eta^2 n_Q^2 \| \mathbf{y}_Q - \mathbf{u}_Q(k) \|_2^2.$$

With these estimates at hand, we are ready to prove the induction hypothesis.

$$\|\mathbf{y}_Q - \mathbf{u}_Q(k+1)\|_2^2$$

$$=\|\mathbf{y}_Q - \mathbf{u}_Q(k) - (\mathbf{u}_Q(k+1) - \mathbf{u}_Q(k))\|_2^2$$

$$=\|\mathbf{y}_Q - \mathbf{u}_Q(k)\|_2^2 - 2\left(\mathbf{y}_Q - \mathbf{u}_Q(k)\right)^\top \left(\mathbf{u}_Q(k+1) - \mathbf{u}_Q(k)\right) + \|\mathbf{u}_Q(k+1) - \mathbf{u}_Q(k)\|_2^2$$

$$=\|\mathbf{y}_Q - \mathbf{u}_Q(k)\|_2^2 - 2\eta\left(\mathbf{y}_Q - \mathbf{u}_Q(k)\right)^\top \mathbf{H}_Q(k)\left(\mathbf{y}_Q - \mathbf{u}_Q(k)\right)$$

$$+2\eta\left(\mathbf{y}_Q - \mathbf{u}_Q(k)\right)^\top \mathbf{H}_Q(k)^\perp \left(\mathbf{y}_Q - \mathbf{u}_Q(k)\right) - 2\left(\mathbf{y}_Q - \mathbf{u}_Q(k)\right)^\top \mathbf{I}_2$$

$$+\|\mathbf{u}_Q(k+1) - \mathbf{u}_Q(k)\|_2^2$$

$$\leq(1 - \eta\lambda_Q + \frac{2C\eta n_Q R}{\delta} + \frac{2C\eta n_Q^{3/2} R}{\delta} + \eta^2 n_Q^2)\|\mathbf{y}_Q - \mathbf{u}_Q(k)\|_2^2$$

$$\leq(1 - \frac{\eta\lambda_Q}{2})\|\mathbf{y}_Q - \mathbf{u}_Q(k)\|_2^2.$$

The third equality we used the decomposition of $\mathbf{u}(k+1) - \mathbf{u}(k)$. The first inequality we used the Lemma 2, the bound on the step size, the bound on $\mathbf{I}_2$, the bound on $\|\mathbf{H}(k)^\perp\|_2$ and the bound on $\|\mathbf{u}(k+1) - \mathbf{u}(k)\|_2^2$. The last inequality we used the bound of the step size and the bound of $R$. Therefore Condition 1 holds for $k' = k + 1$. Now by induction, we prove Theorem 3. $\qquad\square$

Similar to the analysis of lemma 3, we can show the change of $\mathbf{Z}_Q(k)$ and $\mathbf{H}_Q(k)$ is negligible under the conditions of sufficiently large $m$.

**Lemma 6.** *Under the same conditions as Theorem 3 in the transferring process, with probabilility no less than $1 - \delta$,*

$$\|\mathbf{Z}_Q(0) - \mathbf{Z}_Q(k)\|_F = O\left(\frac{n_P^{3/2} n_Q^{5/4}}{\sqrt{m^{\frac{1}{2}} \lambda_P^2 \lambda_Q \kappa^2 \delta^{3/2}}}\right)$$

$$\|\mathbf{H}_Q(0) - \mathbf{H}_Q(k)\|_F = O\left(\frac{n_P^3 n_Q^{7/2}}{\sqrt{m}\lambda_P^2 \lambda_Q \kappa^2 \delta^{\frac{3}{2}}}\right)$$

This is a direct corollary of lemma 4 and can be proved by the same techniques as lemma 3. Now we can continue to analyze the trajectory of transferring $\mathbf{W}(P)$ to the target dataset by dividing the activations in to the two categories as in the proof of Theorem 3.

$$\mathbf{u}_Q(k+1) - \mathbf{u}_Q(k) = -\eta\mathbf{H}_Q(k)(\mathbf{u}_Q(k) - \mathbf{y}_Q) + \boldsymbol{\epsilon}_3(k), \tag{34}$$

$$\|\boldsymbol{\epsilon}_3(k)\|_2 = \sum_{i=1}^{n_Q} \frac{\eta}{m} \sum_{j=1}^{n_Q} \mathbf{x}_{Q,i}^\top \mathbf{x}_{Q,j}\left(u_{Q,j} - y_{Q,j}\right) \sum_{r \in \overline{S}_i} \mathbb{I}\left\{\mathbf{w}_{Q,r}(k)^\top \mathbf{x}_{Q,i} \geq 0, \mathbf{w}_{Q,r}(k)^\top \mathbf{x}_{Q,j} \geq 0\right\} + I_2^i$$

$$\leq \sum_{i=1}^{n_Q} \frac{2\eta\sqrt{n_Q}}{m}|\overline{S}_i|\|\mathbf{u}_Q(k) - \mathbf{y}_Q\|_2,$$

where $|\overline{S}_i| \leq \frac{Cmn_Q R}{\delta} = O\left(\frac{\sqrt{m}n_P^3 n_Q^{\frac{3}{2}}}{\lambda_P^2 \lambda_Q \delta^{\frac{3}{2}}}\right), \forall r \in [m], \forall k \geq 0$. Then substituting $\mathbf{H}_Q(k)$ with $\mathbf{H}_Q^\infty$ using the bound on $\|\mathbf{H}_Q^\infty - \mathbf{H}_Q(k)\|_F$, we further have,

$$\mathbf{u}_Q(k+1) - \mathbf{u}_Q(k) = -\eta\mathbf{H}_Q^\infty(\mathbf{u}_Q(k) - \mathbf{y}_Q) + \boldsymbol{\epsilon}_3(k) + \boldsymbol{\zeta}(k), \tag{35}$$

$$\|\boldsymbol{\zeta}(k)\|_2 \leq \eta\|\mathbf{H}_Q(0) - \mathbf{H}_Q(k)\|_F\|\mathbf{u}_Q(k) - \mathbf{y}_Q\|_2$$

$$\leq \eta\left(1 - \frac{\eta\lambda_Q}{4}\right)^{k-1} \cdot O\left(\frac{n_P\sqrt{n_Q}}{\lambda_P}\right) \cdot O\left(\frac{n_P^3 n_Q^{7/2}}{\sqrt{m}\lambda_P^2 \lambda_Q \kappa^2 \delta^{\frac{3}{2}}}\right), \tag{36}$$

where the second inequality holds with Theorem 3. Taking sum over each iteration,

$$\mathbf{u}_Q(k) - \mathbf{y}_Q = \left(\mathbf{I} - \eta\mathbf{H}_Q^\infty\right)^k (\mathbf{u}_Q(P) - \mathbf{y}_Q) + \sum_{t=0}^{k-1} \left(\mathbf{I} - \eta\mathbf{H}_Q^\infty\right)^k \left(\boldsymbol{\zeta}(k-1-t) + \boldsymbol{\epsilon}_3(k-1-t)\right). \tag{37}$$

$$\|\mathbf{e}(k)\|_2 = \|\sum_{t=0}^{k-1} \left(\mathbf{I} - \eta\mathbf{H}_Q^\infty\right)^k \left(\boldsymbol{\zeta}(k-1-t) + \boldsymbol{\epsilon}_3(k-1-t)\right)\|_2$$

$$\leq k\left(1 - \frac{\eta\lambda_Q}{4}\right)^{k-1} \cdot O\left(\frac{\eta n_P^3 n_Q^{7/2}}{\sqrt{m}\lambda_P^2\lambda_Q\kappa^2\delta^{\frac{3}{2}}}\right) \cdot O\left(\frac{n_P\sqrt{n_Q}}{\lambda_P}\right) = O\left(\frac{n_P^4 n_Q^4}{\sqrt{m}\lambda_P^3\lambda_Q^2\kappa^2\delta^{\frac{3}{2}}}\right), \tag{38}$$

where we notice $\max_{k>0} k\left(1 - \frac{\eta\lambda_Q}{4}\right)^{k-1} = O\left(\frac{1}{\eta\lambda_Q}\right)$. Then in the $K$ iterations of transferring to the target dataset,

$$\text{vec}(\mathbf{W}_Q(K)) - \text{vec}(\mathbf{W}(P))$$

$$= \sum_{k=0}^{K-1} \text{vec}(\mathbf{W}_Q(k+1)) - \text{vec}(\mathbf{W}_Q(k))$$

$$= -\eta\sum_{k=0}^{K-1} \mathbf{Z}_Q(k)(\mathbf{u}_Q(k) - \mathbf{y}_Q)$$

$$= \sum_{k=0}^{K-1} \eta\mathbf{Z}_Q(k)(\mathbf{I} - \eta\mathbf{H}_Q^\infty)^k(\mathbf{y}_Q - \mathbf{u}_Q(P)) - \sum_{k=0}^{K-1} \eta\mathbf{Z}_Q(k)\mathbf{e}(k) \tag{39}$$

$$= \sum_{k=0}^{K-1} \eta\mathbf{Z}_Q(0)(\mathbf{I} - \eta\mathbf{H}_Q^\infty)^k(\mathbf{y}_Q - \mathbf{u}_Q(P))$$

$$+ \sum_{k=0}^{K-1} \eta(\mathbf{Z}_Q(k) - \mathbf{Z}_Q(0))(\mathbf{I} - \eta\mathbf{H}_Q^\infty)^k(\mathbf{y}_Q - \mathbf{u}_Q(P)) - \sum_{k=0}^{K-1} \eta\mathbf{Z}_Q(k)\mathbf{e}(k).$$

The first term is the primary part, while the second and the third are considered perturbations and could be controlled using lemma 6 and equation 38.

$$\|\sum_{k=0}^{K-1} \eta(\mathbf{Z}_Q(k) - \mathbf{Z}_Q(0))(\mathbf{I} - \eta\mathbf{H}_Q^\infty)^k(\mathbf{y}_Q - \mathbf{u}_Q(P))\|_2 = O\left(\frac{n_P^{5/2}n_Q^{3/4}}{\sqrt{m^{1/2}\lambda_P^4\lambda_Q^3\kappa^2\delta^{3/2}}}\right), \tag{40}$$

since $\|\mathbf{Z}_Q(k) - \mathbf{Z}_Q(0)\|_F$ is bounded, the maximum eigenvalue of $\mathbf{H}_Q^\infty$ is $\lambda_Q^{-1}$.

$$\|\sum_{k=0}^{K-1} \eta\mathbf{Z}_Q(k)\mathbf{e}(k)\|_2$$

$$= \|\sum_{k=0}^{K-1} \eta\mathbf{Z}_Q(k)\sum_{t=0}^{k-1} \left(\mathbf{I} - \eta\mathbf{H}_Q^\infty\right)^k \left(\boldsymbol{\zeta}(k-1-t) + \boldsymbol{\epsilon}_3(k-1-t)\right)\|_2$$

$$\leq \eta\sqrt{n_Q} \leq k\left(1 - \frac{\eta\lambda_Q}{4}\right)^{k-1} \cdot O\left(\frac{\eta n_P^3 n_Q^{7/2}}{\sqrt{m}\lambda_P^2\lambda_Q\kappa^2\delta^{\frac{3}{2}}}\right) \cdot O\left(\frac{n_P\sqrt{n_Q}}{\lambda_P}\right)$$

$$= \eta\sqrt{n_Q} \cdot O\left(\frac{1}{\eta^2\lambda_Q^2}\right) \cdot O\left(\frac{\eta n_P^3 n_Q^{7/2}}{\sqrt{m}\lambda_P^2\lambda_Q\kappa^2\delta^{\frac{3}{2}}}\right) \cdot O\left(\frac{n_P\sqrt{n_Q}}{\lambda_P}\right) = O\left(\frac{n_P^4 n_Q^{9/2}}{\sqrt{m}\lambda_P^3\lambda_Q^3\kappa^2\delta^{\frac{3}{2}}}\right), \tag{41}$$

where we use $\sum_{k=1}^{K} k\left(1 - \frac{\eta\lambda_Q}{4}\right)^{k-1} = O\left(\frac{1}{\eta^2\lambda_Q^2}\right)$. With these estimations at hand, we are ready to calculate the final results

$$\|\mathbf{W}_Q(K) - \mathbf{W}(P)\|_F^2$$
$$= \|\operatorname{vec}(\mathbf{W}_Q(K)) - \operatorname{vec}(\mathbf{W}(P))\|_2^2$$
$$= \eta^2(\mathbf{y}_Q - \mathbf{u}_Q(P))^\top \sum_{k=0}^{K-1}(\mathbf{I} - \eta\mathbf{H}_Q^\infty)^{k^\top}\mathbf{Z}_Q(0)^\top\mathbf{Z}_Q(0)\sum_{k=0}^{K-1}(\mathbf{I} - \eta\mathbf{H}_Q^\infty)^k(\mathbf{y}_Q - \mathbf{u}_Q(P))$$
$$+ O\left(\frac{n_P^5 n_Q^5}{m^{1/4}\lambda_P^4\lambda_Q^4\kappa^2\delta^{3/2}}\right)$$
$$= (\mathbf{y}_Q - \mathbf{u}_Q(P))^\top\mathbf{H}_Q^{\infty -1}(\mathbf{y}_Q - \mathbf{u}_Q(P)) + O\left(\frac{n_Q\log\frac{n}{\delta}^{1/4}}{m^{1/4}\lambda_Q}\right) + O\left(\frac{n_P^5 n_Q^5}{m^{1/4}\lambda_P^4\lambda_Q^4\kappa^2\delta^{3/2}}\right)$$
$$\tag{42}$$

Using equation 19, we have

$$\mathbf{y}_Q - \mathbf{u}_Q(P)$$
$$= \mathbf{y}_Q - (\mathbf{Z}_Q(0)^\top(\mathbf{Z}_P(0)(\mathbf{H}_P^\infty)^{-1}\mathbf{y}_P + \boldsymbol{\epsilon}) + \boldsymbol{\epsilon}_1 + \boldsymbol{\epsilon}_2) - \mathbf{u}_Q(0)$$
$$= \mathbf{y}_Q - \mathbf{H}_{PQ}^{\infty\top}(\mathbf{H}_P^\infty)^{-1}\mathbf{y}_P + (\mathbf{H}_{PQ}^{\infty\top} - \mathbf{Z}_Q(0)^\top\mathbf{Z}_P(0))(\mathbf{H}_P^\infty)^{-1}\mathbf{y}_P - \mathbf{Z}_Q(0)^\top\boldsymbol{\epsilon} - \boldsymbol{\epsilon}' + \mathbf{u}_Q(0)$$

With lemma 3, we have

$$\|(\mathbf{H}_{PQ}^{\infty\top} - \mathbf{Z}_Q(0)^\top\mathbf{Z}_P(0))\mathbf{H}_P^{\infty -1}\mathbf{y}_P\|_F \leq O\left(\frac{n_P^{\frac{5}{2}}n_Q}{\sqrt{m}\lambda_P^2\kappa\delta^{\frac{3}{2}}}\right) \tag{43}$$

Combine the estimation above with equation 22 and equation 26. We also have $\|\mathbf{u}_Q(0)\| = O\left(\frac{\sqrt{n}_Q\kappa}{\sqrt{\delta}}\right)$.

$$\mathbf{y}_Q - \mathbf{u}_Q(P) = \mathbf{y}_Q - \mathbf{H}_{PQ}^{\infty\top}(\mathbf{H}_P^\infty)^{-1}\mathbf{y}_P + \boldsymbol{\epsilon}_4 \tag{44}$$

$$\|\boldsymbol{\epsilon}_4\|_2 = O\left(\frac{n_P^{\frac{5}{2}}n_Q}{m^{\frac{1}{2}}\lambda_P^2\kappa\delta^{\frac{3}{2}}}\right) + O\left(\frac{n_P n_Q^{\frac{1}{2}}\kappa}{\lambda_P\delta}\right) + O\left(\frac{n_P^2 n_Q^{\frac{1}{2}}}{m^{\frac{1}{4}}\lambda_P^{\frac{3}{2}}\kappa^{\frac{1}{2}}\delta}\right) + O\left(\frac{n_P^2 n_Q^{\frac{1}{2}}}{m^{\frac{1}{2}}\lambda_P^2\kappa\delta^2}\right) + O\left(\frac{n_Q^{\frac{1}{2}}\kappa}{\sqrt{\delta}}\right)$$
$$= O\left(\frac{n_P^2 n_Q}{m^{\frac{1}{4}}\lambda_P^{\frac{3}{2}}\kappa^{\frac{1}{2}}\delta}\right) + O\left(\frac{n_P\sqrt{n_Q}\kappa}{\lambda_P\delta}\right)$$

Substitute equation 44 into equation 42, we have

$$\|\mathbf{W}_Q(K) - \mathbf{W}(P)\|_F^2 = (\mathbf{y}_Q - \mathbf{H}_{PQ}^{\infty\top}\mathbf{H}_P^{\infty -1}\mathbf{y}_P)^\top\mathbf{H}_Q^{\infty -1}(\mathbf{y}_Q - \mathbf{H}_{PQ}^{\infty\top}\mathbf{H}_P^{\infty -1}\mathbf{y}_P)$$
$$+ 2\frac{1}{\lambda_Q}\|\boldsymbol{\epsilon}_4\|_2\|\mathbf{y}_Q - \mathbf{H}_{PQ}^{\infty\top}\mathbf{H}_P^{\infty -1}\mathbf{y}_P\|_2 + \frac{1}{\lambda_Q}\|\boldsymbol{\epsilon}_4\|_2^2$$
$$+ O\left(\frac{n_Q\log\frac{n}{\delta}^{1/4}}{m^{1/4}\lambda_Q}\right) + O\left(\frac{n_P^5 n_Q^5}{m^{1/4}\lambda_P^4\lambda_Q^4\kappa^2\delta^{3/2}}\right) \tag{45}$$
$$= (\mathbf{y}_Q - \mathbf{H}_{PQ}^{\infty\top}\mathbf{H}_P^{\infty -1}\mathbf{y}_P)^\top\mathbf{H}_Q^{\infty -1}(\mathbf{y}_Q - \mathbf{H}_{PQ}^{\infty\top}\mathbf{H}_P^{\infty -1}\mathbf{y}_P)$$
$$+ O\left(\frac{n_P^2 n_Q\kappa}{\lambda_P^2\lambda_Q^2\delta}\right) + O\left(\frac{n_P^5 n_Q^5}{m^{\frac{1}{4}}\lambda_P^4\lambda_Q^4\delta^2\kappa^2}\right),$$

which completes the proof. $\qquad\square$

