# OpenReview forum: "Towards Understanding the Transferability of Deep Representations"
_ICLR.cc/2020/Conference — Reject_

### Official Review · AnonReviewer2 · 2019-10-22
**Official Blind Review #2**

**Rating:** 3

**Review:**

N.B Since this paper exceeds 8 pages, it was held to a higher standard as per the instructions.

The main goal of this paper is to concretely demonstrate some key properties of transfer in Deep Neural Networks through experiment and theory. By transfer, here they specifically mean the ability for pre-trained networks to be 'updated' to new, similar datasets either completely (all parameters are updated while being initialised by the pre-trained network parameters) or partially (all but the last few layers are kept constant at pre-trained parameter values). There are of course other ways of carrying out transfer learning, but this paper focusses on these methods.They attempt to assess the viability of such a process in its improvements to generalisation and improvements to the loss landscape. In addition, the authors attempt to assess when this type of transfer is viable. The majority of the paper focus on experimental results, while the final 2 pages present some theoretical work that explains those results.

I believe that this work is well motivated. As the paper suggests, there have been several advances in the use of transfer learning that showcase its benefits. That being said, there is a lack of work that systematically tries to explain why these benefits are seen, and how we can better make use of them. This paper tries to fill that gap.

However, while it goes a way in trying to do so, I am not convinced that this paper sufficiently addresses what it sets out to do. This is why I recommend a weak reject, and a summary of my reasons for this are as follows:

1) Section 3: In this section, the authors try to show that transferred networks tend to have better generalisation when the dataset being transferred to is similar to that the network was pre-trained on (ImageNet dataset). The results are shown in Table 1, and they assess this by showing that for new datasets that are more visually similar to the ImageNet dataset, the generalisation error is lower (e.g the Webcam dataset shows lower gen. error than the Stanford Car dataset). I believe that the authors are saying that they are 'visually' similar. However, the ImageNet dataset has subsets that are similar to the Webcam dataset (geological formation), and the Cars dataset (wheeled vehicle). As such, while the goal of the experiment is interesting, it is not clear how interpretable the results are, nor the validity of the conclusions raised.

In addition to this, I would have liked to see the generalisation error of a randomly initialised network for each of the datasets. This would have been an interesting control to see whether the pre-training does indeed improve generalisation.

Further, the authors use the Frobenius norm between the original pre-trained parameters and the final parameters as a measure of how much knowledge is preserved. I am not convinced that this is a sufficiently representative measure of this. I think the extent to which knowledge is preserved is indicated by how well the new, transferred network performs at old tasks from the original dataset. Simply measuring a distance between the parameters doesn't show this. Also consider the fact that it isn't true that networks with parameters a fixed distance away from the original parameters will have similar behaviour.

Figure 3 isn't mentioned anywhere in the text!

2) Section 4: Here, the authors show that pretrained networks lead to flatter, smoother loss landscapes when compared to randomly initialising. This is shown in Figure 5 and Figure 6 mainly. Figure 5 depicts the loss landscapes, and directly shows what this section is claiming. Figure 6 further solidifies this claim by showing that the change in loss at each gradient update is smaller when compared to a randomly initialised network. That being said, the experimental details of this setup is quite sketchy; what dataset is Figure 5 and 6 transferring to, having been pre-trained with ImageNet? Has this been tested between multiple different datasets, multiple times, to show that the conclusions are consistent? Further, it looks like Figure 5a was taken from another paper, why?

3) Section 5: This section tries to answer the question of when transfer learning (as defined by this paper) is viable. Section 5.1 was quite difficult for me to read because I could not understand how the experimental setup described in the text. For example, Section 5.1 says that the network was pretrained on the MNIST dataset and transferred to the SVHN dataset, whereas Figure 8b states the complete opposite. If I assume that the text is correct, the generalisation error in Figure 8b is very difficult to read. In addition to this, I am not sure what the norm of phi-phi(0) is. I also still have my reservations to the use of the norm between parameters, as mentioned above.

In the the section 'Varying Labels with fixed input', the authors mention the use of Caltech-101 and Webcam data, but this isn't mentioned in Figure 8, instead, it mentioned CUB-200, which isn't mentioned in the text. They also mention conclusions from experiments using the Food-101 and Places datasets, but don't show these results anywhere.

Section 5.1 asks important questions, but the authors haven't shown results that can properly answer them.

That being said, Section 5.2 shows the very interesting result that pre-training after a certain number of epochs starts showing diminishing returns in terms of performance of the transferred network.


**Experience Assessment:**

I have read many papers in this area.

**Review Assessment: Checking Correctness Of Derivations And Theory:**

I assessed the sensibility of the derivations and theory.

**Review Assessment: Checking Correctness Of Experiments:**

I assessed the sensibility of the experiments.

**Review Assessment: Thoroughness In Paper Reading:**

I read the paper thoroughly.

---

> ### Author Response · Authors · 2019-11-10
> **Response to AnonReviewer #2 (Part 2)**
>
> 4.	About the figures in Section 5.1
>
> We apologize for the presentation issues in this section, and have fixed these problems in the revision.
>
> - The networks for the digit experiment are pretrained on various versions of MNIST and transferred to SVHN. The scaling of the generalization error is corrected in the revision and is now easy to read. \psi refers to the convolutional kernel in LeNet.
>
> - The “Varying Labels with fixed input” experiments are carried out on transferring from Caltech-101 to Webcam. This figure is now presented in the revision.
>
> - The description of transferring from ImageNet and Places to MIT indoors and CUB200 are presented in Figure 8(c)-(d).
>
> Thanks again for the constructive reviews, which were sincerely appreciated and carefully addressed in the revision. We hope these one-to-one responses are the right answers to your questions.
>
> --
> [1] Peter L Bartlett, Dylan J Foster, and Matus J Telgarsky. Spectrally-normalized margin bounds for neural networks. NIPS, 2017.
> [2] Sanjeev Arora, Rong Ge, Behnam Neyshabur, and Yi Zhang. Stronger generalization bounds for deep nets via a compression approach. arXiv preprint arXiv:1802.05296, 2018.
> [3] Colin Wei, and Tengyu Ma, Data-dependent Sample Complexity of Deep Neural Networks via Lipschitz Augmentation, arXiv preprint arXiv:1905.03684, 2019.
> [4] James Kirkpatrick, Razvan Pascanu, Neil Rabinowitz, Joel Veness, Guillaume Desjardins, Andrei A. Rusu, Kieran Milan, John Quan, Tiago Ramalho, Agnieszka Grabska-Barwinska, Demis Hassabis, Claudia Clopath, Dharshan Kumaran, and Raia Hadsell, Overcoming catastrophic forgetting in neural networks, PNAS 2017 114 (13) 3521-3526, 2017
> [5] Michalis K. Titsias, Jonathan Schwarz, Alexander G. de G. Matthews, Razvan Pascanu, Yee Whye The, Functional Regularisation for Continual Learning, arXiv preprint arXiv:1901.11356, 2019
> [6] Shibani Santurkar, Dimitris Tsipras, Andrew Ilyas, and Aleksander Madry. How does batch normalization help optimization? NeurIPS, 2018.

---

> ### Author Response · Authors · 2019-11-10
> **Response to AnonReviewer #2 (Part 1)**
>
> We really appreciate your insightful comments. All reviewers unanimously believe that this work is dedicated to an interesting and important open problem: understanding the transferability of deep representations. We apologize for the clarity issues in question. To fully address your concerns, we respond to the questions below and summarize them into a minor revision. We will appreciate further advice and discussions based on the revision.
>
> 1.	On the results of Section 3.
>
> The “similarity” here is not the “visual similarity” between data. As we have pointed out in Section 5, what really matters is the similarity between the nature of tasks, i.e. both images and labels matter.
>
> Note that in the sense of granularity, ImageNet, Webcam and Caltech are general classification tasks for mainly coarse-grained objects, while CUB, Stanford Cars are fine-grained classification tasks. In this sense, Webcam and Caltech are more similar to ImageNet, while CUB and Stanford Cars are dissimilar.
>
> We provide the results of training from scratch below, which are also added to Section B.3 (appendix) in the revision. The decreased percentage is calculated by dividing the error reduced in fine-tuning with the error of training from scratch.
> _____________________________
> Dataset       test error  decreased percentage  1/sqrt(n) \|W-W(0)\|_F
> Webcam            26.39		98.29%                               3.54
> Stanford Cars   46.25		48.75%                               5.95
> Caltech-101       22.14           79.36%                               2.34
> CUB-200             52.70          60.15%                                3.90
>
> Comparing the results of training from scratch, the improvement of generalization error is consistent with our observation.
>
> 2.	Quantifying how the knowledge is preserved during fine-tuning with Frobenius norm.
>
> - Correctness is guaranteed. A line of works have shown that the Frobenius norm of the deviation is directly related to the generalization error [1][2][3]. As proved in our theoretical analysis, the Frobenius norm is further upper bounded by a term quantifying the relationship between the pretraining and target tasks (See Theorem 2 in Section 6.3). Thus, the Frobenius norm of deviation is a crucial part in transfer learning, in that it behaves as the link between generalization error and pretrained-target relationship.
>
> - We agree that how well the transferred network performs at old tasks from the original dataset is also a reasonable measure of knowledge preservation, but it is a more suitable measure for “catastrophic forgetting” that was extensively studied in those problems [4][5]. Our measure is more suitable for transfer learning, with attention focused on how much useful knowledge can be preserved to boost the target tasks.
>
> - Also note that the norm is an upper bound of the generalization error, which means small norm of deviation guarantees good transfer performance with high probability, but parameters with a fixed distance does not necessarily perform similarly. For example, if fine-tuning to different target datasets results in models with similar norm of deviation, these models will perform differently, but the generalization error bound will be similar. These results correspond well to our theoretical analysis (Section 6.3), in that the final accuracy will be influenced both by the norm of deviation and the intrinsic property of the target tasks.
>
> 3.	About experiments in Figures 5 and 6.
>
> Experiments in Figure 5 and 6 are carried out on multiple datasets with multiple trials. Figure 5 (b) (c) is carried out on Stanford Cars. Figure 6 is carried out on CUB. Due to the space limitation, we only report results on one dataset in the main paper, and defer results on other datasets to the Section B.1 and B.2 (appendix) in the revision.
>
> Experiments in Figure 6 are inspired by [6], a well-known technique for analyzing Batch Normalization, and we follow its protocols. We add detailed implementation of this experiments to the Section A (appendix) in the revision.
>
> Figure 5a was taken from (He et al. 2018) with proper citation. To remove any possible concern, it is now replaced with another figure to show that transferring from a proper dataset indeed accelerates convergence.

---

### Official Review · AnonReviewer1 · 2019-10-22
**Official Blind Review #1**

**Rating:** 6

**Review:**

The paper gives an extensive empirical and somewhat limited theoretical analysis for the transferability in DNNs. It is shown that transferred models tend to have flatter minima with improved Lipschitzness in the loss function when good choices of pretraining are made.

- The paper is well written and well-organized. Notations and claims are clear.

- This paper presents an interesting line of research, that in my opinion, would be interesting to many researchers in the field, and could attract many follow up works.

- Empirical analysis in sections 3 and 4 are interesting, and give good sense of generalization and optimization landscape. Analyzing the Frobenius norm of the deviation between fine-tuned network and fixed network is reasonable.

- The theoretical analysis seems like a good start, but it is not sufficient in general. There seems to be gap between the network architectures used in empirical evaluations and the theoretical results. However, analyzing transferability is an important topic that needs to be evaluated more. This paper presents interesting new steps towards that goal. That being said, I would be interested to see theoretical results for more general cases alongside experiments on different types of applications.

- Overall, the paper presents a novel approach for evaluating transferability, that I think would be interesting to many researchers in this field. The theoretical results are still limited, and should be investigated more.

**Experience Assessment:**

I have read many papers in this area.

**Review Assessment: Checking Correctness Of Derivations And Theory:**

I assessed the sensibility of the derivations and theory.

**Review Assessment: Checking Correctness Of Experiments:**

I assessed the sensibility of the experiments.

**Review Assessment: Thoroughness In Paper Reading:**

I read the paper at least twice and used my best judgement in assessing the paper.

---

> ### Author Response · Authors · 2019-11-10
> **Response to AnonReviewer #1**
>
> We really appreciate your insightful comments! All reviewers unanimously believe that this work is dedicated to an interesting and important open problem: understanding the transferability of deep representations.
>
> Regarding the theoretical results, we recognize there is still gap between our analysis and realistic networks.
>
> - Analysis of transfer learning relies heavily on the analysis of standard supervised learning. Exact analysis for realistic architectures remains an open problem for standard deep neural networks, let alone the scenario of transfer learning.
>
> - Current over-parametrization based techniques are very popular and implicative of future development on multilayer and other settings, so we analyze the property of transfer learning based on over-parametrization. To our best knowledge, this work provides the first theoretical analysis for transfer learning (fine-tuning) in deep neural networks.
>
> We will delve into the theoretical property of realistic deep networks in our future work.

---

### Official Review · AnonReviewer4 · 2019-11-03
**Official Blind Review #4**

**Rating:** 3

**Review:**

I thank the authors for the clarifications and the modifications to the paper. However, I still lean towards rejection. While the authors provided detailed explanations on some of my points here, most of these are still not in the paper, so the reader would still probably be confused. There are new figures in the appendix, but they are not referred to from the text. The paper feels like a hastily compiled collection of weakly related, somewhat anecdotal and not necessarily clearly explained and motivated experiments. While I believe there is interesting content, with the current presentation style it is really difficult to digest. As an advice, I think it may be better to not squeeze in all the results into one paper, but rather focus on some aspects and analyze them really well, with clear explanation, motivation, and preferably with demonstrating practical gains that result from the analysis - not just hypothetical, but supported by experiments.

---

The paper studies transfer of representations learned by deep networks across datasets and tasks. Namely, the paper analzes the standard setup with pre-training on one dataset and further fine-tuning on a different  dataset. Results include both experiments and theory. On the experimental side, the focus is mainly on TODO. On the theory side, an analysis of transfer in two-layer fully connected networks, based on the ICML-2019 work of Arora et al., is proposed.

I lean towards rejecting the paper, since the presentation and the technical quality are somewhat substandard. This is mainly based on evaluation of the experimental results, since I am not an expert in this subfield of theory and was therefore not able to check the statements and the proofs thoroughly. On the presentation side, many details are missing, which often makes understanding difficult and will likely lead to the results not being reproducible. On the experiments side, the issue is that they are quite anecdotal and have multiple confusing or questionable details, as specified below.

Pros:
1) An interesting topic of trying to understand generalization and transfer in deep learning
2) Multiple types of experiments, including visualizations of loss landscapes at convregence and at initialization, plots of Hessian eigenvalues, measuring the deviation of the weights from their initial values, measuring the variance of the gradients of the weights, measuring the transfer between different datasets, measuring the transfer performance depending on the durarion of pre-training.
3) Theoretical analysis. As mentioned, I cannot really judge about the quality of the results.

Cons:
1) The presentation is quite confusing.
1a) The paper includes many different experiments as well as theory, and it is not very clear how these all come together and what message do they give to the reader. The paper states at one point that it "may stimulate further algorithmic advances", and it would be great if there was a bit more elaboration on this.
1b) Experimental methodology is not presented in the main paper and not referred to. Some of it is described in the appendix, but also not too detailed, for instance the duration of ResNet training is not clear, the details of loss landscape visualization are confusing (for instance, the phrase "... i.e. 0.1× gradient and a total of 200 × 200 grids"),
1c) The paper is over 9 pages, which is more than the recommended length of 8 pages.
1d) Scaling in Figure 8(b) is quite suboptimal, it is impossible to read the test accuracy results.
1e) Minor issues:
 - No color-code in Figure 3 (unclear what values do the colors correspond to) and it does not seem to be referred to in the text.
 - Page 5: "predictive" -> "predictable"?
 - Page 6 "While pretraining on sufficiently large datasets..." - I do not think the experiments tell anything about the dependence of the effect on the size of the dataset, so this phrasing is not justified

2) I found many of the experiments confusing or unconvincing. This is partially affected by the aforementioned issues with presentation.
2a) In Table 1 and Figure 2, it is unclear if difference in generalization between the dataesets is due to similarity to ImageNet (by the way, ImageNet is only mentioned in the caption of Table 1, but not in the text) or due to the inherent properties of the datasets (perhaps some are naturally more difficult or prone to overfitting). I think numbers for training from scratch would be helpful for disambiguating the two cases.
2b) It is unclear why is the norm difference normalized by the square root of the number of target examples. This choice is not justified and it seems it can very noticeably affect the results in counter-intuitive way. For instance, if I understand right, if one duplicates each example in the training set, these values will change, which seems somewhat counter-intuitive. Would it make more sense to normalize by the initial norm of the weights?
2c) In Figure 4, it is unclear if the permutations of the weights are accounted for. The output of a deep network is invariant under a wide variety of weight permutations, and it is natural that networks trained from different random initializations may converge to weights permuted in different ways. In order to meaningfully compare the weight vectors of these networks, one would have to first "align" them, which is a non-trivial task. Another issue I have with the results on Figure 4 is that I don't find them all that counter-intuitive: it seems natural that the weights stay relatively close to the pre-trained network when fine-tuned for that (paritally because the aforementioned permutation symmetry is already largely broken during pre-trianing).
2d) It is unclear which downstream task is analyzed in Figure 6. Moreover, the plot seems to mix two factors: magnitue of the gradient and the smoothness of the loss landscape. Would it not make more sense to make fixed-size steps in the direction of the gradient? Moreover, I am wondering if the difference between the loss values is simply because the overall loss magnitude is larger for the randomly initialized network?
2e) A few parts of subsection 5.1 are confusing. It is not clear where does the interpretation of Figure 8 follow from (perhaps partially because figure 8(b) is partially difficult to read). \psi and W are not defined. What does it move that the weight matrices "show no improvement"? Where are the results on Caltech-101 and Webcam? Why is Food-101 mentioned in the text, but CUB-200 shown in the plots?

**Experience Assessment:**

I have read many papers in this area.

**Review Assessment: Checking Correctness Of Derivations And Theory:**

I did not assess the derivations or theory.

**Review Assessment: Checking Correctness Of Experiments:**

I carefully checked the experiments.

**Review Assessment: Thoroughness In Paper Reading:**

I read the paper at least twice and used my best judgement in assessing the paper.

---

> ### Author Response · Authors · 2019-11-10
> **Response to AnonReviewer #4 (Part 2)**
>
> 2b) Why is the Frobenius norm normalized by the square root of the number of target samples?
>
> This normalization is in line with the generalization error in statistical learning theory [3]. The generalization error always contains the inverse of the square root of the sample number. In practice, even for the same dataset, different number of training samples will lead to different deviation from initialization. That is reasonable since the model has to deviate more to fit more samples. We confirm the validity of this theoretical bound by providing additional results of fine-tuning on CUB-200 in the table below.
>      _____________________________
>      Sample number   500     1000     2000    4000    5994 (full)
> 	\|W-W(0)\|_F     55.1      76.5    112.4   160.7     184.4
>
> The concern on duplication of examples seems a misunderstanding. First, in most generalization theories, the data should not be degenerate, i.e. there are no two identical samples [4]. Second, the training samples in real-world applications are sampled i.i.d. from a continuous distribution. And the probability of sampling one data point twice is 0.
>
> 2c) Regarding the permutation symmetry of the weights in Figure 4.
>
> We totally agree with the insightful points raised by the reviewers. In fact, our results in Figure 4 correspond well to these points. More specifically:
>
> - The permutation symmetry of the weights. We show that, even fine-tuning for the same target dataset, based on models pretrained from different random initializations will converge to very different local minima.
>
> - Permutation symmetry broken with pretraining. We show that, the fine-tuned weights stay relatively close to the pretrained model even when fine-tuned to different datasets.
>
> However, the key points we want to uncover here are none of the above. What we focus on is that, given the above facts, how to understand the transferability of deep representations. Our insights are:
>
> - Due to the permutation symmetry, it is of little sense to quantify the preserved knowledge by the norm of deviation, if not conditioned on a particular pretrained model. Hence both our empirical and theoretical analyses are conditioned on a given pretrained model, such that the norm of deviation is informative to the generalization error bound.
>
> - We show that pretrained representations can implicitly restrict the fine-tuned weight matrices to stay near the pretrained weights. Since the pretrained dataset is sufficiently large and of high-quality, transferring their representations will lead to flatter minima located in large flat basins. This will finally help the generalization performance.
>
> 2d) The experiment in Figure 6.
>
> This experiment follows the same protocol as [5], a well-known technique for analyzing Batch Normalization. It is carried out on CUB-200, a classification task.
>
> - The magnitude of gradient is considered in both [5] and our paper since the smoothness of the loss landscape is a local property. Gradient is the proper scale for analyzing the smoothness.
>
> - We believe fixing step size along the gradient is also a reasonable scenario. To reflect on your advice, we further add results on Stanford-cars with fixed step size along the gradient in Section B.2 (appendix) in the revision. Results are similar for both scenarios.
>
> - The overall loss magnitude has little to do with the loss variation, since initially the loss is the same (log(#class) for cross entropy), and both converge to less than 1e-3.
>
> 2e) About the figures in Section 5.1.
>
> We apologize for the improper scaling and arrangement of those figures. The problems are now fixed in the revision. \psi refers to the convolutional kernel in LeNet. W denotes the weight of the fully-connected layer. The results of transferring from Caltech-101 to Webcam is now provided in Figure 8.2 in the revision.
>
> Thank you again for the constructive reviews, which were very helpful for the revision. And the time and detail put into these reviews were sincerely appreciated.
>
> --
> [1] Xuhong Li, Yves Grandvalet, and Franck Davoine. Explicit inductive bias for transfer learning with convolutional networks. ICML, 2018
> [2] Hao Li, Zheng Xu, Gavin Taylor, Christoph Studer, and Tom Goldstein. Visualizing the loss landscape of neural nets. NeurIPS, 2018
> [3] Mehryar Mohri, Afshin Rostamizadeh, and Ameet Talwalkar. Foundations of machine learning. 2012.
> [4] Simon S. Du, Xiyu Zhai, Barnabas Poczos, and Aarti Singh. Gradient descent provably optimizes over-parameterized neural networks. ICLR, 2019.
> [5] Shibani Santurkar, Dimitris Tsipras, Andrew Ilyas, and Aleksander Madry. How does batch normalization help optimization? NeurIPS, 2018.

---

> ### Author Response · Authors · 2019-11-10
> **Response to AnonReviewer #4 (Part 1)**
>
> Many thanks for your helpful and insightful comments! All reviewers unanimously believe that this work is dedicated to an interesting and important open problem: understanding the transferability of deep representations. Please find our responses to your concerns below. We also attach a minor revision to reflect these responses, and provide the codes to guarantee reproducibility. We will be interested in any further feedback and discussions.
>
> 1a) Elaboration on how this work “may stimulate further algorithmic advances”.
>
> This paper tries to understand the transferability of deep representations from both empirical and theoretical perspectives, and sheds insights on the feasibility of transfer. Basically, the benefit of pretrained representations is two-fold: better generalization and faster optimization, both of which indicate the directions of future fine-tuning algorithms.
>
> - Generalization. Our empirical and theoretical results on generalization error (Sections 3 & 6.3) indicate that the knowledge preserved, as quantified by the norm of the deviation, is determined by the intrinsic similarity between pretraining and target tasks. Note that, the norm of deviation is the consequence, and the similarity between tasks is the cause. Hence, a direct control of this norm term by regularization [1] is not necessarily helpful for drawing the tasks close, especially for dissimilar target tasks where negative transfer could happen. In this sense, we should rethink previous methods that control the norm term explicitly. Further algorithms for fine-tuning should consider implicit regularization to restrict the networks from escaping the flat region of pretrained landscape. And more importantly, seeking pretrained models with larger flat regions to enable safe transfer.
>
> - Optimization. Our empirical and theoretical results on optimization (Sections 4 & 6.2) uncover the major cause of accelerated convergence – improved loss landscapes. However, current optimization methods (e.g. SGD with momentum) for fine-tuning is simply the same as those of training from scratch. Future optimization methods for fine-tuning should harness the landscape information with specific statistics reflecting the similarity between pretraining and target tasks.
>
> - Feasibility. Our results in Section 5.2 reveal an interesting issue of current pretrained models: while pretraining, instead of better performance on the pretraining dataset, a more realistic goal is the transferability of the pretrained models (e.g. less over-fitting, larger flat region). Algorithms for pretraining should take this point into consideration.
>
> 1b) On the experimental methodology and details.
>
> We add all necessary details of implementation in Section A (appendix) in the revision. In particular, the code is provided to reproduce all results.
>
> For fine-tuning, we train ResNet-50 on each dataset for 200 epochs. The visualization method is the same as filter normalization [2]. The resolution of each visualization image is 200 * 200, and the step size is fixed for each figure. Other implementation details can also be found in Section A (appendix).
>
> 1d) and 1e)
>
> We rescale Figure 8 in the revision, so the results are now easy to read. Also, we have fixed several minor issues you raise in the revision.
>
> 2a) About the results of training from scratch in Table 1 and Figure 2.
>
> We definitely agree that the generalization error is also related to the intrinsic property of the dataset itself. Since the focus of this work is transfer learning (fine-tuning), we mainly provide results related to the similarity of tasks.
>
> In fact, the conclusion corresponds well with our theoretical analysis. In Section 6.3, we have already pointed out that for some fine-grained dataset, fine-tuning from another dataset may be less effective, since the target data is “singular”, i.e. the target task itself is very difficult.
>
> To this end, we further provide the results of training from scratch below and also in Table 2 in Section B.3 (appendix) in the revision. The decreased percentage is calculated by dividing the error reduced in fine-tuning with the error of training from scratch.
> _____________________________
> Dataset           test error  decreased percentage  1/sqrt(n) \|W-W(0)\|_F
> Webcam             26.39		         98.29%                      3.54
> Stanford Cars    46.25		        48.75%                       5.95
> Caltech-101       22.14                   79.36%                       2.34
> CUB-200             52.70                   60.15%                       3.90
>
> Comparing the results of training from scratch, the improvement of generalization error is still consistent with our observation.

---

### Decision · Program_Chairs · 2019-12-19

**Decision:**

Reject

**Comment:**

This paper studies the transfer of representations learned by deep neural networks across various datasets and tasks when the network is pre-trained on some dataset and subsequently fine-tuned on the target dataset. The authors theoretically analyse two-layer fully connected networks and provide an extensive empirical evaluation arguing that the loss landscape of  appropriately pre-trained networks is easier to optimise (improved Lipschitzness).
Understanding the transferability of representations is an important problem and the reviewers appreciated some aspects of the extensive empirical evaluation and the initial theoretical investigation. However, we feel that the manuscript needs a major revision and that there is not enough empirical evidence to support the stated conclusions. As a result, I will recommend rejecting this paper in the current form.
Nevertheless, as the problem is extremely important I encourage the authors to improve the clarity and provide more convincing arguments towards the stated conclusions by addressing the issues raised during the discussion phase.